# High-throughput protein characterization by complementation using DNA barcoded fragment libraries

Bradley W Biggs [1], Morgan N Price [1], Dexter Lai [2], Jasmine Escobedo [2], Yuridia Fortanel [2], Yolanda Y Huang [1], Kyoungmin Kim [2], Valentine V Trotter [1], Jennifer V Kuehl [1], Lauren M Lui [1], Romy Chakraborty [1], Adam M Deutschbauer [1,3] & Adam P Arkin [1,2 ✉]

## Abstract

**Our ability to predict, control, or design biological function is fundamentally limited by poorly annotated gene function. This can be particularly challenging in non-model systems. Accordingly, there is motivation for new high-throughput methods for accurate functional annotation. Here, we used complementation of auxotrophs and DNA barcode sequencing (Coaux-Seq) to enable high-throughput characterization of protein function. Fragment libraries from eleven genetically diverse bacteria were tested in twenty different auxotrophic strains of *Escherichia coli* to identify genes that complement missing biochemical activity. We recovered 41% of expected hits, with effectiveness ranging per source genome, and observed success even with distant *E. coli* relatives like *Bacillus subtilis* and *Bacteroides thetaiotaomicron*. Coaux-Seq provided the first experimental validation for 53 proteins, of which 11 are less than 40% identical to an experimentally characterized protein. Among the unexpected function identified was a sulfate uptake transporter, an O-succinylhomoserine sulfhydrylase for methionine synthesis, and an aminotransferase. We also identified instances of cross-feeding wherein protein overexpression and nearby non-auxotrophic strains enabled growth. Altogether, Coaux-Seq's utility is demonstrated, with future applications in ecology, health, and engineering.**

**Keywords** DNA Barcoding; Functional Genomics; High-throughput Characterization; Protein Annotation
**Subject Category** Microbiology, Virology & Host Pathogen Interaction

## Introduction

Understanding the core metabolic functions of an organism is a critical step toward predicting its behavior and rationally manipulating function (Bordbar et al, 2014; Frioux et al, 2020; Widder et al, 2016). While a substantial amount of core metabolism is conserved, our ability to accurately annotate even well-known function is limited (Schnoes et al, 2009). For example, a recent study found that nearly one-third of a diverse set of 127 bacteria were erroneously predicted to be auxotrophic for an amino acid (Price, 2023). Poor annotation at this level undermines approaches such as genome-scale metabolic modeling, where gene annotation is foundational and errors introduced at the annotation step contribute significant uncertainty (Ankrah et al, 2021; Bernstein et al, 2021). As various applications progress towards non-model organisms, the existence of isozymes, alternative biosynthetic pathways, and low amino acid identity homologs will continue to present a challenge. Failure to accurately identify core metabolic pathways will lead to misassignment of auxotrophies, misunderstanding of adaptive physiologies, and inferences of community dependencies that likely do not exist. Therefore, there is a need for new and high-throughput approaches to provide experimental evidence to accurately annotate these central pathways.

To this end, our laboratory has developed a suite of high-throughput functional genomics methods based on DNA barcoding and parallel fitness profiling, including transposon mutagenesis libraries (RB-TnSeq) (Price et al, 2018; Wetmore et al, 2015), dual-barcoded *E. coli* genomic fragment shotgun expression libraries (Dub-Seq) (Mutalik et al, 2019), and single-barcoded overexpression library screening in the anaerobe *Bacteroides thetaiotaomicron* (Boba-Seq) (Huang et al, 2022), along with the development of CRISPRi tools for similar applications (Qi et al, 2013; Rishi et al, 2020). Here, we extend the DNA barcoding framework in functional genomics by employing shotgun expression libraries of diverse bacteria for the **co**mplementation of **aux**otrophs (Coaux-Seq, where the biochemical function is "coaxed" from genetic material). By utilizing long-read sequencing to link genomic fragments to 20 nucleotide DNA-barcodes, we take advantage of facile and cost-effective barcode sequencing (BarSeq) to repeatedly assay genomic fragment libraries in different genetic contexts to test for the ability of contained gene material to encode protein(s) that complement a missing biochemical function in *E. coli*. Thus,

[1]Environmental Genomics and Systems Biology Division, Lawrence Berkeley National Laboratory, Berkeley, CA 94720, USA. [2]Department of Bioengineering, University of California-Berkeley, Berkeley, CA 94720, USA. [3]Department of Plant and Microbial Biology, University of California-Berkeley, Berkeley, CA 94720, USA. ✉E-mail: aparkin@lbl.gov

we link genes from diverse bacterial genomes to known function. For this study, we generated 11 diverse bacterial genomic fragment libraries and tested these libraries in 20 different genetic knockout contexts. Beyond providing the first experimental evidence to validate the predicted biochemical function of 42 enzymes, we identified the function of eight homologously divergent enzymes. Further, we identified unexpected function in three enzymes, including a sulfate uptake transporter from the TauE family, an O-succinylhomoserine sulfhydrylase for methionine synthesis (MetZ), and an aminotransferase.

# Results

To ascertain the genetic range of material capable of being successfully utilized in our proposed workflow, specifically in the context of *E. coli* expression, we sampled from genetically diverse bacteria. This included both gram-positive (e.g., *Bacillus subtilis*) and gram-negative (e.g., *Pseudomonas fluorescens*) strains, three different phyla (Bacillota, Pseudomonadota, and Bacteroidota), and 11 different genera. Classes represented include alphaproteobacteria (e.g., *Sphingomonas koreensis*), betaproteobacteria (e.g., *Xylophilus sp.*), gammaproteobacteria (e.g., *Lysobacter sp.*), bacteroidia (e.g., *Bacteriodes thetaiotaomicron*), sphingobacteriia (*Pedobacter* sp.), and bacilli (e.g., *B. subtilis*). The full set of 11 strains can be found in Table 1 and Fig. 1A, and summary information on the generated libraries can be found in Tables 1 and 2, Figs. EV1 and EV2 (Datasets EV1 and EV2), and Appendix Figs. S1–11. Several strains were obtained through recent isolations from the Oak Ridge Reservation, and their isolation conditions are described in detail in the methods. If not already available, the genome for each strain was sequenced and added to NCBI (full set of accession numbers in "Methods").

The full experimental workflow for Coaux-Seq can be found in Fig. 1A–I. Briefly, extracted genomes are sheared by sonication to ~3 kb fragment size (Appendix Fig. S12 for an example gel), end-repaired, phosphorylated, and blunt-end ligated into a barcoded expression vector. For each of the 11 libraries, long-read sequencing

(PacBio) was used to link the ligated genome fragment to the unique barcode of its vector, following the library mapping workflow of Boba-Seq (Huang et al, 2022). By establishing this link between the fragment and barcode once, subsequent experiments only required more cost-effective barcode sequencing (BarSeq) to identify the genome fragment(s) contained. Mapped libraries could then be transformed, either individually or in groups, into different auxotrophic strains of *E. coli* and cultivated under selective conditions. For strains that grew under selection, the DNA could be extracted and sequenced by BarSeq.

Transformations were initially recovered in rich medium to determine transformation efficiency, both with respect to colony-forming units (spotting assays) and with respect to the number of barcodes transformed (BarSeq). Rich medium recovered transformations were archived and could later be assayed under selective conditions. Selection was carried out in a defined minimal medium (M9) with 1% glucose as the sole carbon source. A synthetic, TetR repressed, Tet promoter was provided upstream of the genomic fragment insert to enable synthetic expression. Appendix Fig. S13 confirms inducible mRFP expression for this vector. The *E. coli* single gene knockout strains into which the libraries were transformed were auxotrophic, specifically unable to grow in M9 medium with glucose as a sole carbon source. Where growth was observed, colonies were scraped from the selective plates into sterile 1× PBS, pelleted, and plasmid miniprepped. Miniprepped plasmids were used as a template for BarSeq to determine which fragments were selected through this process.

Each of the 11 libraries were transformed individually into four different auxotrophic backgrounds (ΔaroA, ΔproB, ΔmetB, ΔthrB) and as a combination of all 11 libraries into each of the 20 auxotrophic strains (ΔaroA, ΔproB, ΔmetB, ΔthrB, ΔcysA, ΔargG, ΔleuA, ΔtrpA, ΔserA, ΔhisC, ΔmetE, ΔilvD, ΔpheA, ΔpurE, ΔproA, ΔcysH, ΔhisG, ΔaroE, ΔpyrD, Δppc) (full information of auxotrophic strains in Appendix Table S1, full list of all library transformations in Appendix Table S2, and full list of all selections in Appendix Table S3). This list of knockout strains was chosen based on the aforementioned criteria of being auxotrophic in minimal medium with glucose as a sole carbon source and to cover

**Table 1. Summary table on genome fragment libraries generated.**

| Strain | cfus | BarSeq | PB read # | PB size | Gene # | Gene Cov. |
|---|---|---|---|---|---|---|
| *Escherichia coli* BW25113 | 6.6E + 04 | 56,500 | 954,803 | 2628 | 4233 | 98% |
| *Sphingomonas koreensis* JSS26 (DSMZ 15582) | 9.8E + 04 | 34,200 | 352,158 | 2424 | 4167 | 91% |
| *Bacillus subtilis* 168 | 8.2E + 04 | 68,700 | 506,000 | 2300 | 4240 | 90% |
| *Bacteroides thetaiotaomicron* VPI-5482 | 5.6E + 04 | 52,000 | 497,338 | 2074 | 4682 | 55% |
| *Pseudomonas fluorescens* FW300-N2E2 | 4.5E + 04 | 39,700 | 476,967 | 2695 | 6014 | 97% |
| *Lysobacter sp.* FW306-1B-D06B | 5.2E + 04 | 18,900 | 85,984 | 1976 | 3818 | 65% |
| *Xylophilus sp.* GW821-FHT01B05 | 5.9E + 04 | 13,500 | 435,755 | 1981 | 5311 | 51% |
| *Rhodanobacter denitrificans* FW104-10B01 | 8.5E + 04 | 46,200 | 355,863 | 2108 | 3534 | 82% |
| *Rhodoferax sp.* GW822-FHT02A01 | 8.0E + 04 | 20,900 | 569,040 | 2075 | 5121 | 81% |
| *Pedobacter sp.* FW305-3-2-15-E-R2A2 | 6.5E + 04 | 31,200 | 550,499 | 2257 | 6256 | 43% |
| *Acidovorax sp.* FHTAMBA | 9.8E + 04 | 52,700 | 336,044 | 2192 | 4344 | 90% |

"Cfus" column represents colony-forming units based on plating after transformation on LB chloramphenicol agar plates (selective for plasmid transformation). "BarSeq" represents the estimated number of unique barcodes as identified by BarSeq. "PB read #" is the total number of PacBio sequencing reads for that library. "PB size" is the average size in base pairs of the insert as identified by PacBio sequencing. "Gene #" is the total number of genes in the reference genome. "Gene Cov." is the percentage of genes on the genome covered by the generated fragment library.

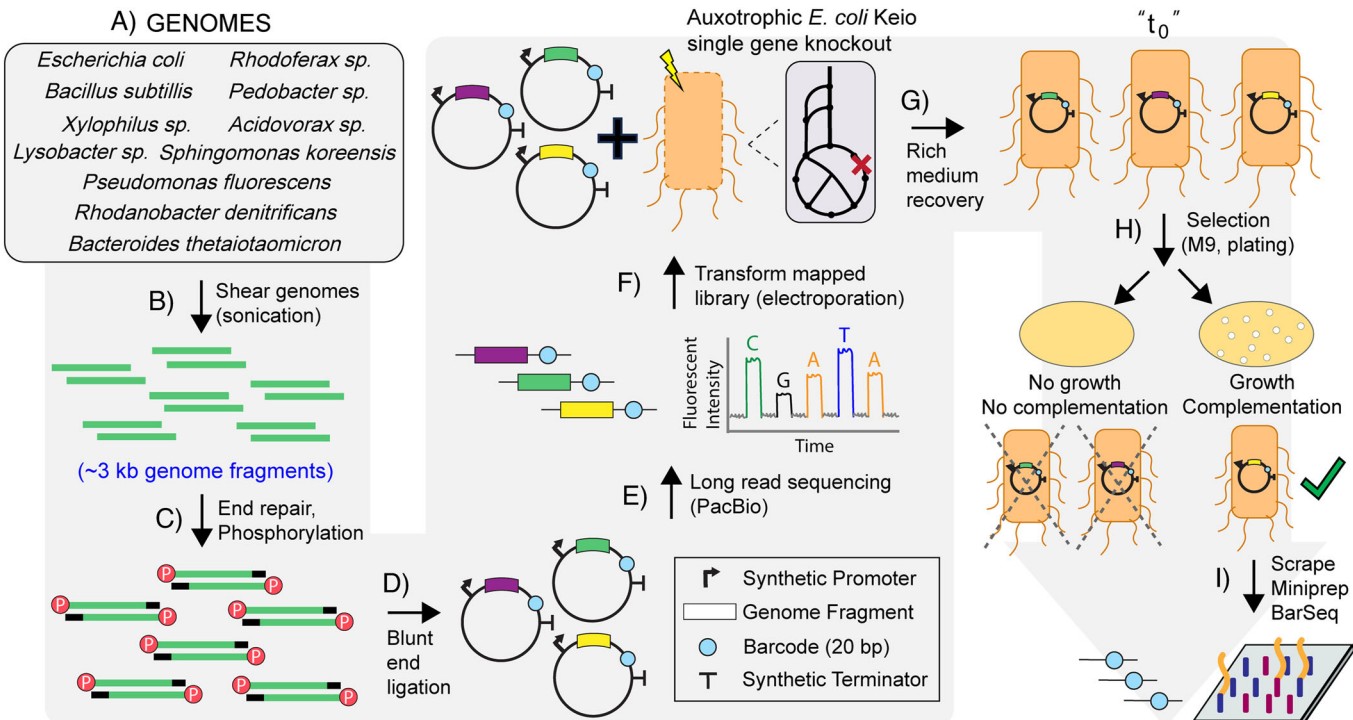

**Figure 1. Coaux-Seq full workflow.**

(A) Organisms are chosen and genomes extracted. (B) These genomes are sheared by sonication to ~3 kb fragment size. (C) Fragments are end-repaired and phosphorylated. (D) Blunt-end ligation is used to ligate the repaired and phosphorylated fragments into the barcoded (random PCR generated 20 nucleotide segments) expression vector. (E) Long-read (PacBio) sequencing is used to map the fragments to the barcodes. (F) The libraries are transformed into a given auxotrophic strain. (G) Transformed strains are recovered in rich medium (SOC, LB). (H) Selection is conducted in a minimal (M9, glucose) solid medium. (I) After selection, colonies are scraped, miniprepped, and BarSeq analysis is run.

**Table 2. Summary of PacBio sequencing of fragment libraries, genes per fragment, and fragments covering each gene.**

| Genome | Genome Size (kb) | #mapped Fragments | Avg. Size (kb) | Std. dev. Size (kb) | Genes per fragment | | | | | Fragments per gene | | | | |
|---|---|---|---|---|---|---|---|---|---|---|---|---|---|---|
| | | | | | Avg. | 0 | 1 | 2 | >=3 | Avg. | 0 | 1 | 2 | >=3 |
| *Acidovorax sp.* FHTAMBA | 4752 | 18,608 | 2.2 | 0.6 | 1.3 | 25% | 38% | 25% | 12% | 5.3 | 11% | 7% | 9% | 73% |
| *Bacillus subtilis* 168 | 4216 | 41,949 | 2.4 | 0.6 | 1.4 | 27% | 33% | 25% | 15% | 13.1 | 12% | 5% | 3% | 80% |
| *Bacteroides thetaiotaomicron* VPI-5482 | 6293 | 24,087 | 2.1 | 0.6 | 0.8 | 46% | 33% | 14% | 7% | 4.1 | 48% | 8% | 7% | 38% |
| *Escherichia coli* BW25113 | 4631 | 25,772 | 2.7 | 0.7 | 1.6 | 18% | 32% | 28% | 22% | 9.3 | 9% | 2% | 3% | 86% |
| *Lysobacter sp.* FW306-1B-D06B | 4236 | 7609 | 2.1 | 0.6 | 1.1 | 35% | 35% | 20% | 10% | 2.1 | 36% | 16% | 14% | 34% |
| *Pedobacter sp.* FW305-3-2-15-E-R2A2 | 7588 | 8992 | 2.3 | 0.8 | 0.9 | 48% | 28% | 15% | 9% | 1.3 | 59% | 15% | 10% | 17% |
| *Pseudomonas fluorescens* FW300-N2E2 | 6921 | 29,012 | 2.8 | 0.7 | 1.5 | 20% | 32% | 28% | 20% | 7.3 | 6% | 3% | 5% | 86% |
| *Rhodanobacter sp.* FW104-10B01 | 3959 | 16,432 | 2.2 | 0.6 | 1.1 | 33% | 35% | 21% | 10% | 5.1 | 20% | 9% | 9% | 62% |
| *Rhodoferax sp.* GW822-FHT02A01 | 5487 | 11,952 | 2.2 | 0.7 | 1.2 | 31% | 36% | 22% | 12% | 2.7 | 20% | 15% | 18% | 47% |
| *Sphingomonas koreensis* JSS26 | 4399 | 16,788 | 2.5 | 0.6 | 1.5 | 23% | 32% | 26% | 19% | 5.9 | 11% | 5% | 7% | 77% |
| *Xylophilus sp.* GW821-FHT01B05 | 5833 | 6560 | 1.9 | 0.6 | 1 | 35% | 40% | 18% | 7% | 1.2 | 49% | 18% | 16% | 17% |

Table shows the genome size in kilobases (kb), the number of mapped fragments per library, and the average size of mapped fragments (kb). In addition, a breakdown of how many genes were contained per fragment for each genome and the number of different fragments covering each gene per genome is shown.

a range of chemistries within core metabolism. These activities included transferase (AroA; EC 2.5.1.19), kinase (ProB; EC 2.7.2.11), lyase (MetB: EC 2.5.1.48), synthetase (ArgG; EC 6.3.4.5), dehydratase (IlvD; EC 4.2.1.9), reductase/dehydrogenase (ProA; EC 1.2.1.41), carboxylase (Ppc; EC 4.1.1.31), and transporter

(CysA; EC 7.3.2.2/5). The majority of knockouts came from amino acid biosynthesis (17), with two (purE, pyrD) from nucleotide biosynthesis and one (Ppc) from central metabolism (Appendix Table S1). All experiments were run at 37 °C. Initial experiments were plated at multiple densities and optimized with respect to aTc

induction level (Appendix Fig. S14), with greater induction (transcription) tested to potentially compensate for weak heterologous ribosomal binding site (RBS) activity. From these experiments, 5× aTc (500 ng/mL) was found optimal (Appendix Fig. S14), and subsequent experiments were run at both 1× (100 ng/mL) and 5× aTc, with 1× aTc used as a precaution for possible overexpression toxicity due to a 5× aTc induction level.

Initial library selections were performed on solid medium. This allowed for the maintenance of a diversity of barcodes as measured by BarSeq and prevented consolidation to only a single or few barcodes. When selection was conducted for initial experiments in liquid culture, the top two barcodes accounted for >99% of all reads in all three initial cases tested (Appendix Tables S4–6). While top barcodes corresponded to expected genes, these assays utilizing the individual libraries for *E. coli*, *S. koreensis*, and *P. fluorescens* FW300-N2E2 transformed into Δ*thrB* delivered 1, 1, and 2 of the possible 15, 9, and 17 expected gene fragments, respectively. In contrast, solid medium selection for the same libraries in Δ*thrB* delivered 14, 4, and 10 of the expected gene fragments, indicating a clear bias introduced by liquid selection. Selection on agar plates, however, created a potential for false positives in the form of very small colonies that achieve minimal growth either due to carryover of nutrients even after multiple (3×) washing steps or from nutrients contained in agar by way of impurities. Upon scraping plates in preparation for plasmid miniprep and BarSeq, these faint colonies can contribute a small fraction of cell mass and thus DNA to subsequent experiments, necessitating cutoff criteria for enrichment to delineate successful fragments as defined below. These small faint colonies were observed even in control experiments run with red fluorescent protein (mRFP) in place of a genome fragment. However, false positive colonies did not grow beyond a small translucent state on solid medium. In addition, these false positive colonies did not grow when transferred to liquid medium (M9, 1% glucose with chloramphenicol, kanamycin, and aTc), save a single instance for Δ*pheA* as discussed further below.

## Total library hits

Using this workflow, we identified complementation "hits," meaning fragments containing genes for which the encoded biochemical activity improved the fitness of the auxotrophic *E. coli* in the selective condition (allowed growth). Figure 2 shows an example for TK06_RS12685 from *P. fluorescens* FW300-N2E2 in the context of Δ*hisC*. Highlighted in green are fragments that contain the whole TK06_RS12685 gene. Several fragments containing the whole gene show elevated fitness scores, while neighboring fragments that do not contain TK06_RS12685 do not show improved fitness scores. Fitness was defined as the $\log_2$ change in the relative abundance of a specific barcode in a given experiment. This was computed as the normalized $\log_2$ ratio of the number of reads for the barcode from the experimental sample (scraped up from an agar plate after growth in minimal medium) divided by the number of reads for that barcode from the control sample (used to inoculate the plate). We defined high-confidence hits as fragments that provided significant benefit (fitness >5 and z-like test statistic >4) and either (1) an overlapping fragment provided a significant benefit in that same experiment or (2) the fragment provided a significant benefit in another experiment in the same mutant background (regardless of induction level).

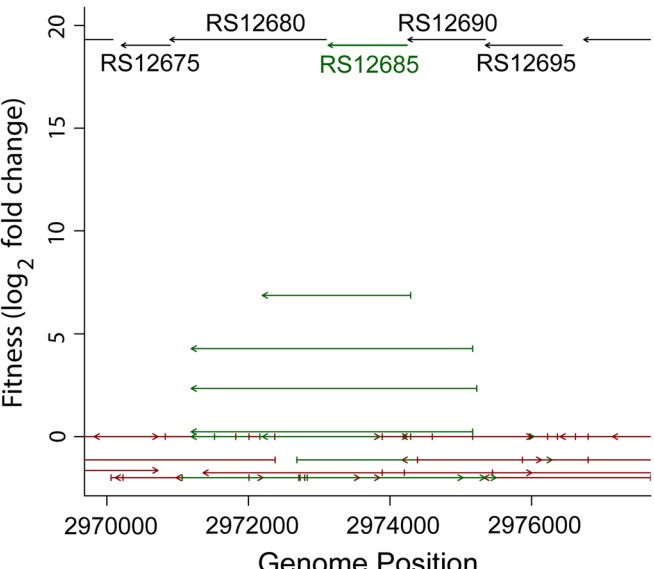

**Figure 2.  Example hit.**

The figure shows TK06_RS12685 from *P. fluorescens* FW300-N2E2 for activity in Δ*hisC*. Fragments that contain all of the gene (TK06_RS12685) are highlighted in green. Fitness values (y axis) represent the average over two experiments. This specific example comes from an experiment where all 11 libraries were transformed together into the knockout background. If a fragment is higher on the y axis, it showed a greater fitness improvement. The top of the plot shows a reference segment of the source genome. Length of the fragments below gives an indication of which genes were included in the given fragment. The arrow on the fragment indicates its directionality within the expression vector. For this case, all of the fragments with fitness >0 are in the same orientation as the reference gene, indicating that they are all in the sense orientation with respect to the synthetic promoter.

Figure 3 (Dataset EV3) shows an overview of the fragments with potential benefits (fitness >4) across all experiments. A dark blue "x" denotes a fragment that provided a significant benefit and the green diamonds indicate a fragment with high fitness that overlapped with another fragment with high fitness. Many of the inserts with high fitness overlap another insert with high fitness (63% if either fitness value is above 5). In addition, many of the inserts with high fitness show a benefit at both inducer concentrations (74% of markers that are above 5 for one axis are above 5 for both).

In total, we identified 838 instances of a fragment in the context a specific knockout background providing a significant fitness benefit in at least one experiment. Of these, 420 cases met the criteria of a high-confidence benefit. Combining the overlapping fragments among the 420 high-confidence hits, 160 regions were identified to provide high-confidence fitness benefits (a summary flowchart is provided in the Discussion section). We associated these regions with genes by considering all genes that were contained within the insert with the highest fitness (averaging across experiments for that genetic background). Considering the 160 regions, 64 are associated with just one open reading frame (ORF), 75 are associated with two or more ORFs (47 with exactly two), and 21 do not contain an entire ORF.

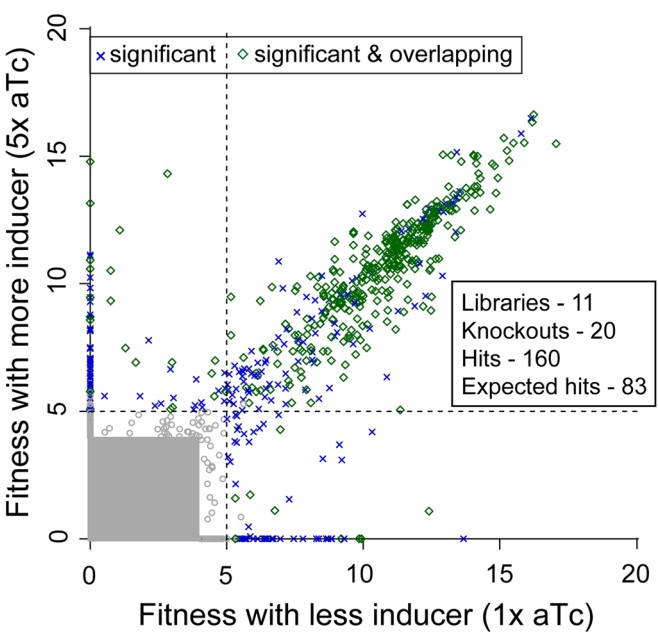

## Comparison of Replicates

Legend: × significant   ◇ significant & overlapping

Libraries - 11
Knockouts - 20
Hits - 160
Expected hits - 83

x-axis: Fitness with less inducer (1× aTc)
y-axis: Fitness with more inducer (5× aTc)

**Figure 3.  Coaux-Seq replicate comparison.**

Each point indicates the fitness values of a specific barcode from two experiments with the same mutant background and the same library (or mixture of libraries), but with different concentrations of the inducer. The fitness value from a lower concentration of inducer is on the x axis. Points are highlighted if the barcode has a statistically significant benefit (fitness >5 and z-like test statistic >4) in either experiment or if the barcode has a significant benefit and overlaps another barcode with a significant benefit. Points that are not statistically significant either fall within the gray box or are not highlighted. Fitness is a $\log_2$ fold change. Most pairs of measurements (over 4 million) have no fitness benefit (both $\log_2$ ratio <4) and lie within the gray box.

To identify causative genes, we first considered genes that lie within these regions and are expected to perform the missing function.

Although we did not perform exact replicate experiments, we repeated many of our experiments at more than one concentration of inducer, including experiments with the mixed library from 11 genomes for 19 of the 20 mutant backgrounds (because of a low number of colonies from the $\Delta proA$ transformation, the 1× and 5× aTc plates were combined). As mentioned, and demonstrated in Fig. 3, barcodes that provided a benefit at one concentration usually provided a benefit at the other concentration. This was almost always true if that barcode's insert overlapped with the insert of another barcode that was significant in the same experiment (see green points in Fig. 3). In particular, if an insert had a significant benefit at the lower concentration, then it had a significant benefit at the higher concentration 88% of the time. For inserts whose benefit was confirmed by overlap, this proportion rose to 98%.

## Expected hits

Based on the known genes of the source genomes and existing annotation tools, it is possible to create a list of enzymes that a priori would be expected to complement our auxotrophic knockout strains even if they have not been experimentally verified. To create

a list of expected hits, we considered all of the knockouts except $\Delta cysA$, as CysA is the ATPase subunit of a 4-component sulfate transporter complex, and its association and functionality within the protein complex may present an independent issue beyond examining for a single enzyme catalytic function. From there, GapMind (Price et al, 2020), an annotation tool for amino acid biosynthesis that relies on experimentally characterized enzymes, was used to identify candidates for amino acid biosynthesis knockouts. For CysH and PyrD, TIGRFAMs (Haft et al, 2003) was used to identify candidates, using HHMer 3.3.1 and the trusted cutoff for the bit score (as provided by the TIGRFAMs curators). For PurE, we searched for homologs of the *E. coli* or *B. subtilis* PurE enzymes, which yielded a single candidate per genome. For Ppc, we only searched for homologs of *E. coli* phosphoenolpyruvate carboxylase, as *B. subtilis* does not have this enzyme. If a candidate enzyme was high-confidence (>40% amino acid identity to an experimentally validated enzyme that had the appropriate function, and where it was the only candidate in the genome), it was included as an expected hit.

There were some special cases that required additional effort, such as instances where there was more than one high-confidence candidate or if the original annotation was vague. For these, we used PaperBLAST (Price and Arkin, 2017), a tool that finds papers about homologs by using a combination of full-text search (EuropePMC (Gou et al, 2015)) and curated resources (Swiss-Prot (Bateman et al, 2023), BRENDA (Chang et al, 2021), EcoCyc (Caspi et al, 2020)), for confirmation. Steps without high-confidence candidates in the genome were manually examined, focusing on the lower-confidence GapMind hits. This yielded two additional candidates. First, AAFF35_21465 from *Pedobacter sp.* FW305-3-2-15-E-R2A2 was identified as a likely AroA. Close homologs of AAFF35_21465 are essential proteins, and it is 45% identical to HMPREF1058_RS13970 from *Phocaeicola vulgatus* CL09T03C04, which is cofit with chorismate synthase (AroC) in RB-TnSeq fitness data (Surya Tripathi, personal communication). As RB-TnSeq utilizes transposon mutagenesis to disrupt genes across a host organism (loss-of-function), correlated fitness often indicates that the two genes have related function (Price et al, 2018). Second, AAGF34_00495 of *Rhodoferax sp.* GW822-FHT02A01 was considered a likely SerA, as it is 68% identical to BPHYT_RS03150 from *Burkholderia phytofirmans* PsJN, which was identified as SerA using RB-TnSeq data (Price et al, 2018). For MetB complementation, GapMind's candidates for MetZ, which can substitute for both MetB and MetC, were also considered. However, these instances were excluded if the source organism was predicted to use O-acetylhomoserine in its methionine synthesis pathway instead of *E. coli*'s native O-succinylhomoserine pathway, as these candidates are unlikely to function appropriately in this background. For Ppc, two diverged homologs were included (LRK54_RS12075 of *Rhodanobacter denitrificans* FW104-10B01 and AAFF32_01485 of *Lysobacter sp.* FW306-1B-D06B). While these homologs are only 36–37% identical to characterized phosphoenolpyruvate carboxylases, they are likely to function as Ppc considering the functional residues are conserved, as identified using SitesBLAST (Price and Arkin, 2022). In addition, AAFF32_01485 from *Lysobacter sp.* FW306-1B-D06B is 58% identical to N515DRAFT_2010 from *Dyella japonica* UNC79MFTsu3.2, which is confirmed to be Ppc by its fitness pattern in RB-TnSeq data (Price et al, 2018). Overall, we identified 203 proteins expected to complement one of the 19 mutants, with 7–14 per mutant background and 14–22 per source genome.

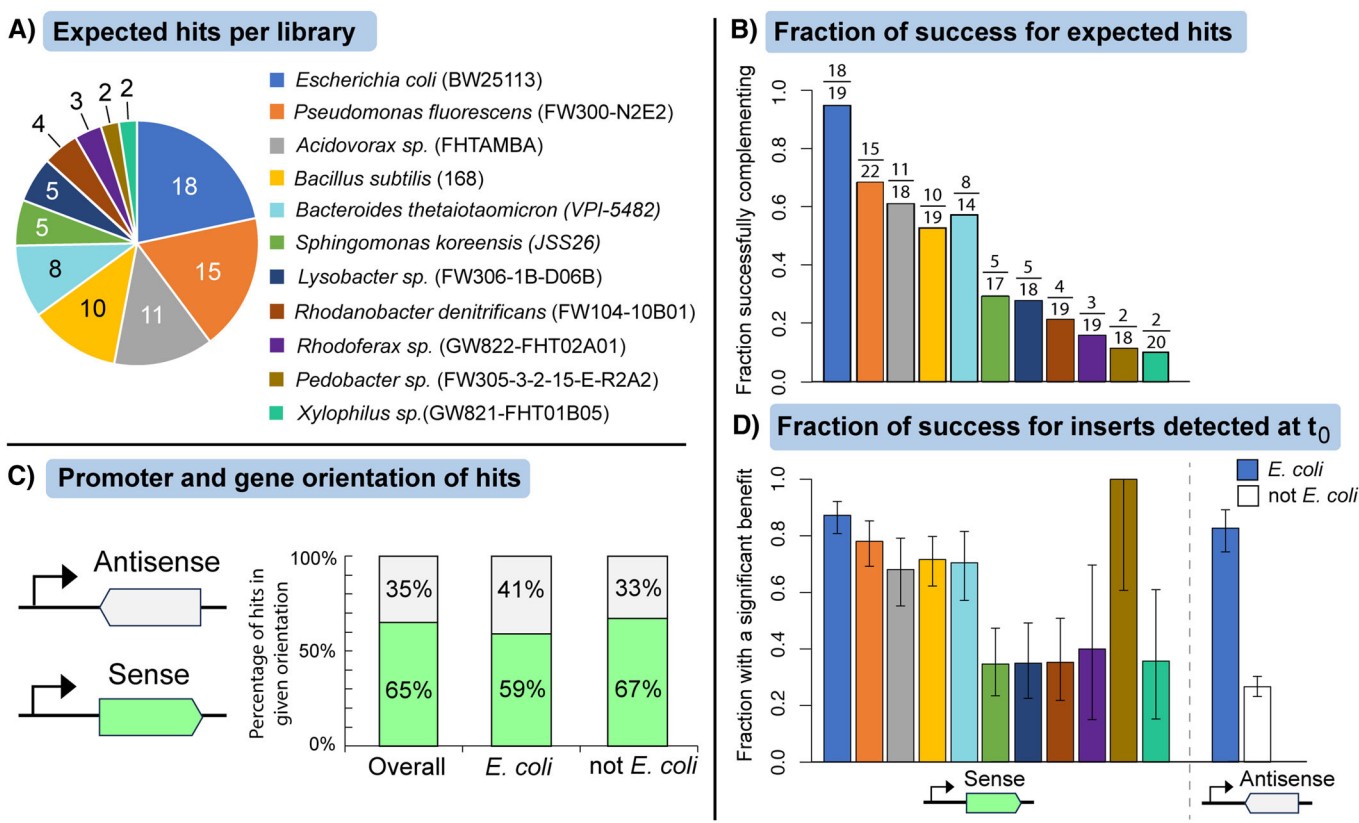

**Figure 4. Successfully recovered expected hits.**

(A) Shows the breakdown of the genome of origin for the recovered expected hits. Strain ordering and color follow the legend in panel (A) throughout the figure. (B) Shows the fraction of the possible complementing genes per genome that were successful. Fractions are shown above. Denominator values are determined by the number of potential complementing genes in the genome for the 19 auxotrophies considered. (C) Shows the breakdown of the orientation of the complementing gene with respect to the synthetic promoter among inserts that have a statistically significant benefit and contain a gene expected to confer a benefit in that experiment. The upper gray portion of the bars represents the antisense orientation, and the lower green portion of the box represents the sense orientation. (D) Shows the fraction of inserts, among those detected at $t_0$ and expected to provide a benefit, that were successful. We show the success rate separately for inserts in the sense (left of vertical dashed line) and antisense (right of vertical dashed line) orientation with respect to the synthetic promoter. Center of error bars is the fraction of inserts that show a significant benefit that were detected at $t_0$ and expected to provide a benefit. Error bars in (D) represent the 90% confidence interval (binomial test).

## Expected hit recovery

We initially assessed if each *E. coli* gene could successfully complement its own loss and be identified in our assay. We observed successful complementation for 19/20 (95%) auxotrophs, including CysA. The only absent complementation was CysH, which both had a low transformation efficiency (few barcodes observed at $t_0$) and a potential complicating factor of futile ATP usage stress (Gillespie et al, 1968). Considering the broader set of 203 expected complementing proteins, 80 were covered by a high-confidence benefit fragment containing a full-length gene and 3 were covered by a nearly full-length gene (41% recovery of expected hits), representing 52% of all of the high-confidence hits. As mentioned, *E. coli* showed the highest recovery rate (95%). *P. fluorescens* FW300-N2E2 showed the second highest recovery rate (68%, Fig. 4A,B; Appendix Table S7). All other libraries showed a lower recovery rate.

In an effort to understand why some expected hits were not recovered, we looked into several possible explanatory effects. First, we checked to see if a fragment containing the expected hit was present in the generated libraries, and a full-length gene was found in the libraries for 75% of the expected hits. Second, we examined whether the expected gene was contained in the library at $t_0$ (meaning right after transformation and rich medium recovery, before selection), to determine if the fragment transformed successfully. Of the remaining 152 genes covered by a fragment, only 132 of them were seen at $t_0$ (summary flowchart provided in the Discussion section). Moreover, as would be expected, inserts that contain a should-be beneficial protein but were not detected in the $t_0$ samples (0 reads for their barcode) are much less likely to show a benefit compared to inserts that are detected. If a fragment contained a complete gene that was expected to provide and was observed at $t_0$, then we observed a benefit 50% of the time. If a fragment contained a complete gene that was expected to provide a benefit but was not observed at $t_0$, then we observed a benefit just 2% of the time, with the 2% likely representing fragments that were transformed at low abundance and missed in the $t_0$ sequencing. Together these effects accounted for roughly 70 of the 120 not recovered hits.

Next, we examined whether relatedness to *E. coli* was explanatory for success. This analysis considered fragment recovery percentage for

fragments detected at $t_0$ and in the sense orientation with respect to the synthetic promoter. We found no clear correlation between fragment recovery success and fragment genome origin relatedness to *E. coli*. For example, one of the most distantly related organisms, *B. subtilis*, showed good recovery of expected complementing genes (72%, Fig. 4D; Dataset EV4). At the same time, the gammaproteobacteria *Lysobacter sp.* FW306-1B-D06B and *R. denitrificans* FW104-10B01 showed poor recovery (35% each), while other gammaproteobacteria (*E. coli* and *P. fluorescens* FW300-N2E2) were among the best with respect to recovery (87% and 78%, respectively). Similarly, the betaproteobacteria *Rhodoferax sp.* GW822-FHT02A01 and *Xylophilus sp.* GW821-FHT01B05 showed poor recovery (40% and 36%), but other betaproteobacteria, including *Acidovorax sp.* FHTAMBA, showed good recovery (68%). While the alphaproteobacteria *S. koreensis* showed poor recovery of expected hits (35%), it was the only representative. All the other genome libraries besides *E. coli* showed between 60 and 70% recovery (Fig. 4D; Dataset EV4).

Next, we considered possible expression effects. Among inserts that were detected in the $t_0$ samples and are expected to provide a benefit, those that have the gene in the same orientation as the synthetic promoter are nearly 2× more likely to demonstrate a benefit (66% vs. 35%). Following, we investigated expression impacts of ribosomal binding sites (RBS). Because of the manner of assembly of the libraries (random shearing), our expression vector contained a synthetic promoter but not a synthetic RBS, as the gene could be located anywhere in the ~3 kb region of the insert fragment and in either orientation. Therefore, the expression of fragment-contained genes depends on native RBS function. We used the OSTIR calculator (Roots et al, 2021) to determine expected expression strength of source genetic material RBSs. The only library with significantly weaker RBS strength on average was *S. koreensis* (mean RBS strength of 1.3, as compared to 2.4 across other libraries; $P = 0.007$, $t$ test with Bonferroni correction for 11 libraries tested), and this library did perform poorly (35% recovery). However, *S. koreensis* contains a relatively high number of leaderless transcripts (Lomsadze et al, 2018), which would inherently deliver a low RBS score, as no RBS would be present, and are unlikely to be expressed in *E. coli* anyway. When considering only inserts that cover the entire gene, were oriented correctly with respect to the synthetic promoter, and were detected in the $t_0$, the mean RBS strength was 2.36 for inserts with significant benefits and 2.37 for those without ($P = 0.89$, $t$ test). Overall, the RBS strength prediction did not correlate with individual protein complementation success ($P = 0.40$, Wilcoxon rank-sum test).

We also looked at genome GC content. Several of the genomes with low success rates have much higher GC content than *E. coli*. Across all inserts that contain an expected beneficial gene, are detected at $t_0$, and in the sense orientation to the synthetic promoter, the median GC content was 61% for successful fragments and 66% for unsuccessful fragments ($P = 0.006$, Wilcoxon rank-sum test). The poorer performance of the higher GC content genomes could have been related to expression via codon usage. However, comparing percentage of rare codons or codon adaptation index (CAI (Sharp and Li, 1987)) for genes that did or did not complement produced no clear takeaways. A modest correlation for CAI ($P = 0.049$, Wilcoxon rank-sum test) was observed but in the opposite direction of expectation, meaning successful genes had a lower codon adaptation score (0.52) compared to unsuccessful genes (0.57), which is likely an artifact. Appendix Fig. S15 provides plots comparing successful and unsuccessful cases

with respect to their GC content, predicted RBS strength, and codon adaptation index. As can be seen, successful and unsuccessful cases have almost completely overlapped values, and thus no obvious trend can be ascertained.

Considering the 83 expected hits recovered, all 11 libraries were represented with at least two hits each, with the distribution favoring *E. coli*, *P. fluorescens* FW300-N2E2, and *B. subtilis* (Fig. 4A). And while RBS strength was not predictive of individual protein's success rate to complement, the RBS strength calculations did show that *B. subtilis* had the highest average score (3.6), which may be due to its low GC content and could contribute to the somewhat unexpected strong performance in *E. coli*. For comparison, *E. coli* had the second highest average score (3.0). Lastly, as expected, a majority of the hits were in the sense orientation with respect to the synthetic promoter (65%, Fig. 4C; Dataset EV4). Excluding the *E. coli* gene fragments, for which natively contained promoters would be expected to perform well in our assays, 67% of hits were in the sense orientation with respect to the synthetic promoter (Fig. 4C).

## Diverged hits

In addition, we categorized ten of the hits as "diverged". Of these, eight are low (<40% identity) homologs, and are included among the expected hits. The additional two we labeled as "other" hits. These cases could have been expected to complete the missing function, but with caveats to high-confidence assignment. Hits were considered "diverged" if they were similar to experimentally characterized proteins that have the missing activity (and not more similar to proteins known to have other functions instead), but are <40% identical on an amino acid sequence basis to any characterized protein with that activity in curated databases (Swiss-Prot (Bateman et al, 2023), MetaCyc (Caspi et al, 2020), BRENDA (Chang et al, 2021), and fitness browser reannotations (Price et al, 2018)). Appendix Table S8 provides a full list of such cases. With respect to the first "other" case, AAGF34_01100 from *Rhodoferax sp.* GW822-FHT02A01 successfully complements *E. coli* Δ*hisC*, while only being 31% identical to hisC Q8R5Q4 of *Thermoanaer-obacter tengcongensis*. While AAGF34_01100 was identified by TIGRfam, GapMind found two additional candidates, one of which (AAGF34_11740) was >50% homologous to a previously characterized hisC and was selected as the "expected hit." At the same time, AAGF34_01100 is found in a histidine biosynthesis operon, so this behavior could be expected. For the second "other" case, *B. subtilis* MetI (MetB-like) was found to complement Δ*metB*, even though *B. subtilis* uses acetylated intermediates and *E. coli* uses succinylated intermediates. This complementation was previously reported to work, even though the enzyme has no detectable activity on O-succinylhomoserine (Auger et al, 2002).

## Identifying novel enzymes and transporters

With respect to the remaining 75 hits that were not associated with an a priori expected complementation of the auxotrophic activity or our diverged set, we chose to follow up with 14 cases for validation (Appendix Table S9). Fitness plots for the associated fragments can be found in Appendix Figs. S16–29 and Fig. 2. For each of these cases, the identified gene fragment was cloned from the host genome into the expression vector exactly as it was found in the library fragment. Following, it was tested directly in the relevant auxotrophic *E. coli*

host, selected on solid medium, inoculated into liquid medium, and the plasmid was miniprepped and sent for Sanger sequencing to confirm complementation and avoid false positives.

Of these 14 cases tested, 6 were found not to complement, 4 were found to benefit by cross-feeding, and 4 were found to successfully complement. Figure 5 (Dataset EV5) shows a breakdown of the fitness scores and frequency of overlap for each of these categories along with the expected, diverged, and untested unexpected hits, and shows that the 6 cases found not to complement trend towards lower fitness scores and lack of overlapping fragments. Cross-feeding and successful cases are described in further detail below. Among the cases that were confirmed to not complement were the putative transporter BSU_32380 of *B. subtilis* in the context of Δ*cysH*, BW25113_RS20525 (*ppc*) of *E. coli* in the context of Δ*proB*, TK06_RS22845 (*ilvD*) of *P. fluorescens* FW300-N2E2 in the context of Δ*proB*, and TK06_RS26365 of *P. fluorescens* FW300-N2E2 in the context of Δ*pyrD*. Likewise, the two putative major facilitator superfamily (MFS) transporters LRK54_RS17455 of *R. denitrificans* FW104-10B01 and TK06_RS20405 of *P. fluorescens* FW300-N2E2 were both found to not complement in the context of Δ*pheA*.

Several additional unexpected hits were found not to complement when individually expressed in the appropriate auxotrophic strain, but among a subset of these we recognized the possibility of a fitness benefit for the contained fragment in the context of a nearby non-auxotrophic strain cross-feeding a needed metabolite. For example, the unsuccessful fragments tested in the context of Δ*ppc* were putative dicarboxylate transporters or symporters. When

## Categories of Hits

**Figure 5. Distribution of fitness scores for different categories of hits.**

Next to the category label is the median fitness value (bold) and the percentage of hits that had a confirming overlapping fragment in the category. As can be seen, hits that were not validated are more likely to have lower fitness values (closer to 5) and are less likely to be confirmed by overlapping fragments. For those that did not validate only one had a single overlapping fragment. To note, the single +14.7 supported-by-overlap hit in the "other unexpected" category is *B. subtilis* ornithine aminotransferase (BSU_40340) complementing Δ*proA*. Both enzymes have the same product, glutamate semi-aldehyde, thus possibly explaining this seeming outlier. Fitness is a log$_2$ fold change.

spotted on a plate proximal to wild-type *E. coli*, strains over-expressing these transporters showed improvement of growth compared to a control (mRFP in the place of a gene fragment) (Appendix Fig. S30). These results indicated that the overexpression of these transporters provided a growth benefit, which occurs likely through improved uptake of dicarboxylic acids. Of possible candidate metabolites, succinate is most likely, as *E. coli* is known to secrete it (Clark, 1989) and it provides an alternate entry point to the TCA cycle. Similarly, overexpression of BT_RS23500 introduced to Δ*trpA* showed a growth benefit in cross-feeding assay, contrasting a negative control (mRFP), and only in the case of spotting with a wild-type *E. coli* (Appendix Fig. S31). BT_RS23500 is 59% identical to TrpB2 from *Thermotoga sp.* (Q9WZ09), an enzyme that forms tryptophan from indole (Hettwer and Sterner, 2002). This suggests that indole secreted from the wild-type *E. coli* (Wang et al, 2001) could be taken up by the auxotrophic strain and converted to tryptophan by BT_RS23500 to alleviate the loss of TrpA's role of cleaving indole-3-glycerol phosphate (IGP) into glyceraldehyde 3-phosphate (GAP) and indole as part of tryptophan biosynthesis. It is somewhat surprising that *E. coli's* native TrpB cannot provide this same benefit, but BT_RS23500 may possess superior kinetics. Its homolog, *Thermotoga* sp. trpB2 (Hettwer and Sterner, 2002) has a K$_M$ for indole below 1 μM, while TrpB from *E. coli* (in the absence of TrpA) has a K$_M$ of 14 μM. Together, these results indicate the importance of considering cross-feeding in analyzing complementation assay results.

The two MFS transporters identified in the context of Δ*pheA* (LRK54_RS17455 of *R. denitrificans* FW104-10B01 and TK06_RS20405 of *P. fluorescens* FW300-N2E2) were also tested for cross-feeding growth benefit. Neither demonstrated a clear cross-feeding benefit, though strains overexpressing these transporters showed greater plate-based growth than the negative control (Appendix Fig. S32). Unexpectedly, negative controls in these assays sometimes grew to a significant amount (Appendix Fig. S32, agar plate on the right in the figure as an example). As previously mentioned, in other knockout contexts, negative controls would sometimes show small faint colonies on solid medium, but did not grow in liquid culture. However, if given extended periods of time (~1 week), negative controls in Δ*pheA* grew in liquid culture. The long-lag time growth was different compared to strains overexpressing truly complementing fragments like TK06_RS12685 of *P. fluorescens* FW300-N2E2 and BT_RS19865 of *B. thetaiotaomicron*, which grew in 1–2 days. These observations led us to examine this behavior further.

To understand what might be occurring, we sequenced the genome of a Δ*pheA* negative control strain that grew to look for a compensating mutation that may facilitate this growth. However, we found no clear candidate mutations that might provide a benefit (Dataset EV6 for all differences between wild-type *E. coli* BW25113 and this strain). PheA is a bifunctional enzyme with both chorismate mutase and prephenate dehydratase activity. Because *E. coli* possesses alternative enzymes for the chorismate mutase activity, Δ*pheA* complementing fragments are generally expected to be carrying out the prephenate dehydratase activity. Interestingly, this activity can occur spontaneously, particularly at low pH (Cerutti and Guroff, 1965; Kishore et al, 1999). Therefore, it is possible that over extended periods of time, non-enzymatically catalyzed reactions are sufficient to provide growth for negative controls. As we consistently allowed multiday windows for growth in confirmation experiments, we observed Δ*pheA* growth when

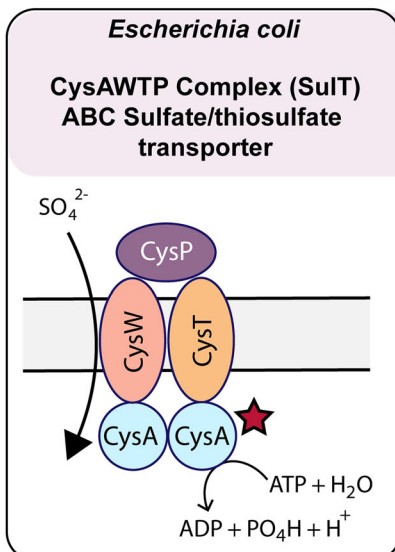
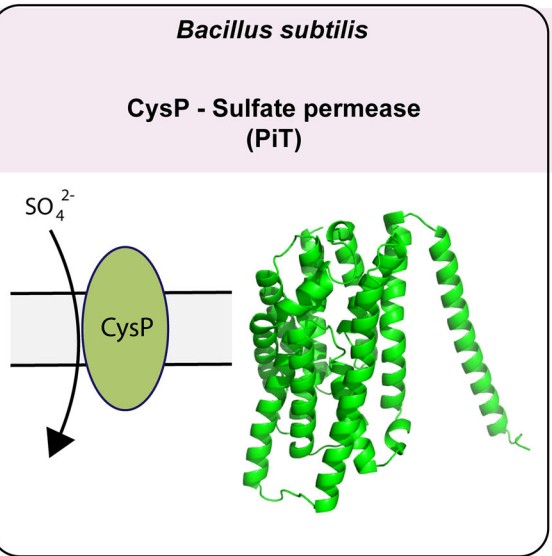
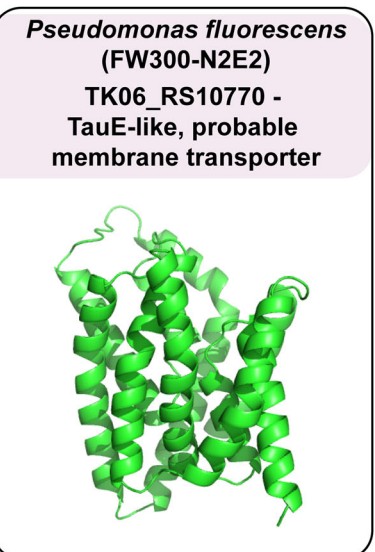

**Figure 6.** Complementation of CysA activity by *Bacillus subtilis* CysP and by novel sulfate transporter TK06_RS10770 from *P. fluorescens* FW300-N2E2. First panel shows the native *E. coli* system.

Second panel shows the function of the sulfate permease (CysP) from *B. subtilis*, along with its AlphaFold predicted structure. The final panel shows the AlphaFold predicted structure for TK06_RS10770 of *P. fluorescens* FW300-N2E2.

others may not have. These results indicate that spontaneous or even slow alternative enzymatic reactions should be taken into account as possible causes for cell growth when analyzing complementation assay data, particularly when extended time periods for growth are allowed (such as for follow up experiments). Nevertheless, our complementation assays successfully identified 8 of the expected 12 PheA proteins (including *E. coli* PheA) as complementing this mutant, so this phenomenon did not entirely undermine the assay.

The four remaining unexpected hits were confirmed to complement. Among the confirmed complementing fragments, two examples came from the context of Δ*cysA*. Sulfur is an essential element for microbial metabolism, and microorganisms, including *E. coli*, utilize sulfate importers to obtain it. *E. coli* employs a sulfate/tungstate uptake transporter (SulT) family complex CysUWA-CysP/sbp, where CysP and sbp are alternate periplasmic substrate-binding proteins (Fig. 6) (Aguilar-Barajas et al, 2011). Within this ABC importer complex, CysA is responsible for ATP energy coupling (Aguilar-Barajas et al, 2011). Accordingly, when testing fragments for complementation in Δ*cysA*, one would expect to identify hits that are CysA homologs. Beyond identifying *E. coli* CysA to complement itself, we only found two other fragments that allowed for *E. coli* growth in the context of Δ*cysA*, neither of which contained a CysA homolog. First, and perhaps as could be expected, we found that CysP from *Bacillus subtilis* replaces the missing functionality to allow for *E. coli* growth. CysP, a sulfate permease from the phosphate inorganic transporter (PiT) family (Mansilla and De Mendoza, 2000), complements the lost activity of CysA by replacing the entire CysUWTP complex function. More interestingly, a previously unannotated transporter, the TauE homolog TK06_RS10770 of *P. fluorescens* FW300-N2E2, was also confirmed to restore *E. coli* Δ*cysA* growth. The TauE family includes a sulfite exporter and a sulfoacetate exporter, but to date has not previously

been linked to sulfate uptake. That stated, TK06_RS10770 is 49% homologous to Ac3H11_578 from *Acidovorax sp.* GW101-3H11, which is an operon with sulfate assimilation genes. Moreover, data from transposon mutants also links Ac3H11_578 to sulfate assimilation. Across 140 RB-TnSeq experiments, the fitness pattern of Ac3H11_578 is most correlated with a subunit of sulfate adenylyltransferase (linear correlation = 0.94) (Price et al, 2018). Combining the complementation data, RB-TnSeq data, and genome context, we concluded that TK06_RS10770 and Ac3H11_578 are sulfate uptake transporters. Interestingly, as this protein does not bear homology to previously characterized sulfate transporters and is in a different family compared to the permease components (CysW and CysT), it is possible that it represents an example of convergent evolution.

A second example of a confirmed hit is that of LRK54_RS05660 of *R. denitrificans* FW104-10B01 found in the context of Δ*metB*. MetB is a cystathionine gamma-synthase that, along with cystathionine beta-lyase (MetC), is involved in the essential two-step process of forming L-homocysteine from L-cysteine and O-succinyl-L-homoserine as part of *E. coli*'s methionine synthesis pathway (Fig. 7). By homology, LRK54_RS05660 is related to both *E. coli* MetB (41%) and *E. coli* MetC (28%), with its AlphaFold structure resembling MetB (Fig. 7). Yet, its closest biochemically-characterized homolog is cystathionine gamma-lyase from *Pseudomonas aeruginosa* (65%) (Pedretti et al, 2024). Moreover, RB-TnSeq data for LRK54_RS05660 does not show auxotrophic phenotypes (Hira Lesea, personal communication) leaving unclear its function. Specifically, RB-TnSeq assays with a pool of mutants in minimal glucose medium found fitness values >0 for this gene, which indicates that mutants of this gene had no growth disadvantage (thus, could not be auxotrophic). As some microbes utilize an alternative pathway from O-succinyl-L-homoserine to L-homocysteine by way a single step with MetZ (Fig. 7), we tested to see if LRK54_RS05660 were able to also complement Δ*metC*, another Keio collection knockout, in

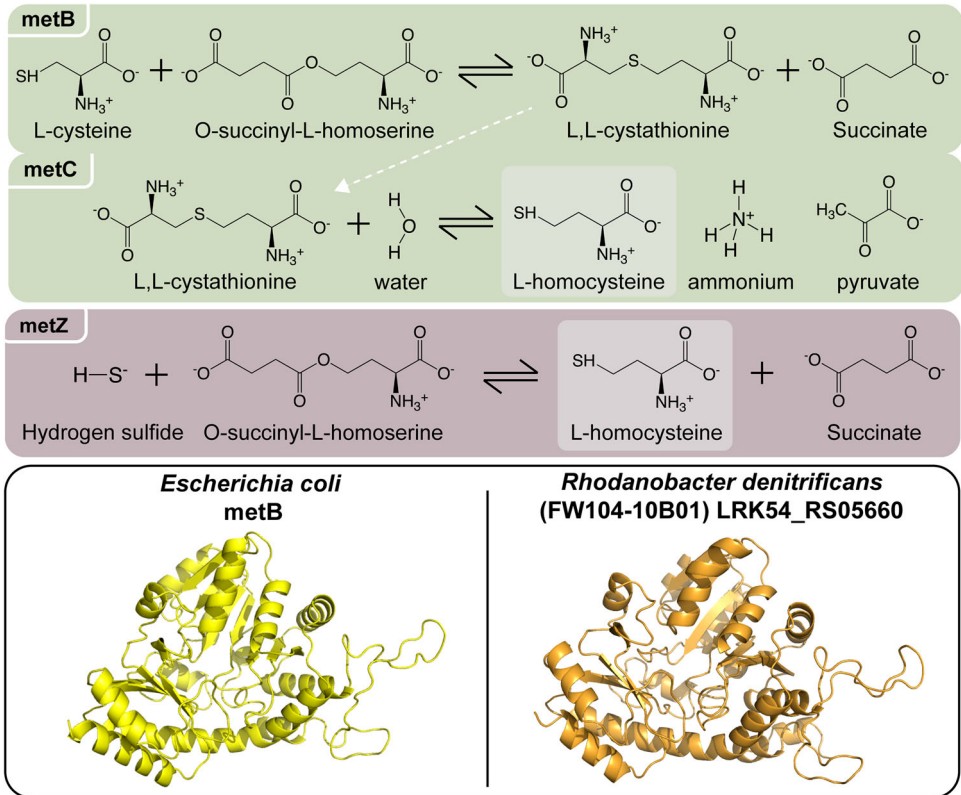

**Figure 7.  Identification of a protein, LRK54_RS05660 of *R. denitrificans* FW104-10B01 that complements Δ*metB* knockout.**

The top of the figure shows the two dominant pathways from homoserine to homocysteine used by bacteria, the two-step (MetBC) and one step (MetZ) pathways. The lower portion of the figure shows crystal structure of *E. coli* MetB and the AlphaFold predicted structure for LRK54_RS05660, which bears structural similarity.

addition to Δ*metB*, which might indicate it was instead a MetZ. LRK54_RS05660 was not found able to complement Δ*metC*.

In contrast, another gene that complements Δ*metB*, AAFF19_12795 of *Acidovorax sp*. FHTAMBA, is predicted as either a methionine gamma-lyase or MetB, and we experimentally found that AAFF19_12795 was able to fill the role of a MetZ. AAFF19_12795 is 90% identical to Ac3H11_2452 from *Acidovorax sp*. GW101-3H11, which is strongly cofit with homoserine O-succinyltransferase (*metA*, $r = 0.98$ (Price et al, 2018)). Interestingly, neither FHTAMBA nor GW101-3H11 has a strong candidate for MetC, and the best candidate in the GW101-3H11 strain (Ac3H11_34) is more similar to methionine gamma-lyases and is not important for growth in minimal media (all RB-TnSeq fitness >0, indicating no disadvantage for mutants of this gene) (Price et al, 2018). Because of the lack of a clear MetC enzyme in these genomes, we hypothesized that both AAFF19_12795 and Ac3H11_2452 were MetZ (Foglino et al, 1995). Our initial experimental evidence of AAFF19_12795 complementing Δ*metB* leaves possible either MetB or MetZ functionality. Therefore, we tested AAFF19_12795 for Δ*metC* complementation, and found that it indeed complemented Δ*metC*, providing evidence that it is likely MetZ. Though this general activity could have been expected because of homology, additional assays were necessary to clarify its protein function. These results indicate the importance of considering possible alternative functions when analyzing complementation assay data.

As a final example, TK06_12685 of *P. fluorescens* FW300-N2E2 was confirmed to complement Δ*hisC* (Fig. 8). TK06_RS12685 exists within a conserved operon containing aromatic amino acid biosynthesis genes for both tyrosine and phenylalanine synthesis, along with SerC and cytidylate kinase, suggesting a possible role related to amino acid biosynthesis. It is 81% identical to PA3165 (HisC2) of *P. aeruginosa*. And while HisC2 is not required for histidine synthesis (Wang et al, 2020), TK06_12685 also has 43% homology to *B. subtilis* HisC, providing further possible connection to this role. Interestingly, TK06_RS12685 also has 55% homology to BPHYT_RS14905 of *Burkholderia phytofirmans* PsJN, which is thought to be a phenylalanine transaminase because it is important for fitness during growth on phenylalanine (Price et al, 2018). Together, this evidence might suggest that, based on sequence similarity, TK06_12685 is a transaminase for phenylalanine or histidinol phosphate. Our data strongly suggests that it can use histidinol phosphate as a substrate, even though this might not be its physiological role. In summary, these three examples show that beyond validating expected function, Coaux-Seq successfully uncovered novel biochemical function.

## Discussion

Coaux-Seq provides another important tool for functional genomics and advancing high-throughput annotation of genetic function. Coaux-Seq builds on existing gain-of-function tools like Dub-seq (Mutalik et al, 2019) and Boba-seq (Huang et al, 2022),

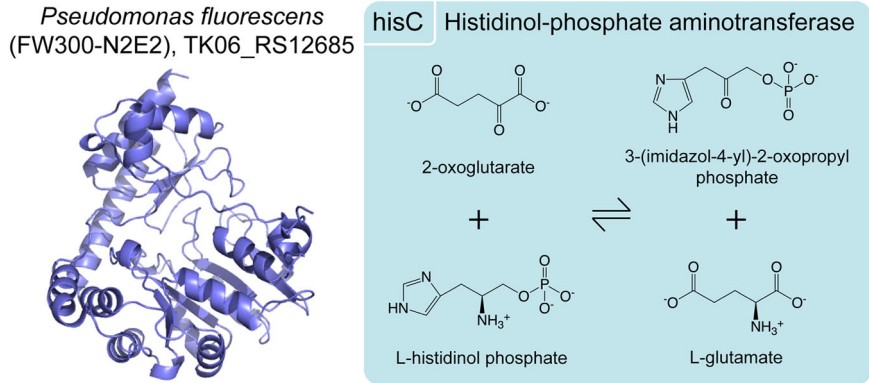

**Figure 8.  HisC activity from TK06_RS12685 of *Pseudomonas fluorescens* FW300-N2E2.**

One the left is the AlphaFold predicted structure of TK06_RS12685. On the right is the activity of HisC from *E. coli* that it is being complemented.

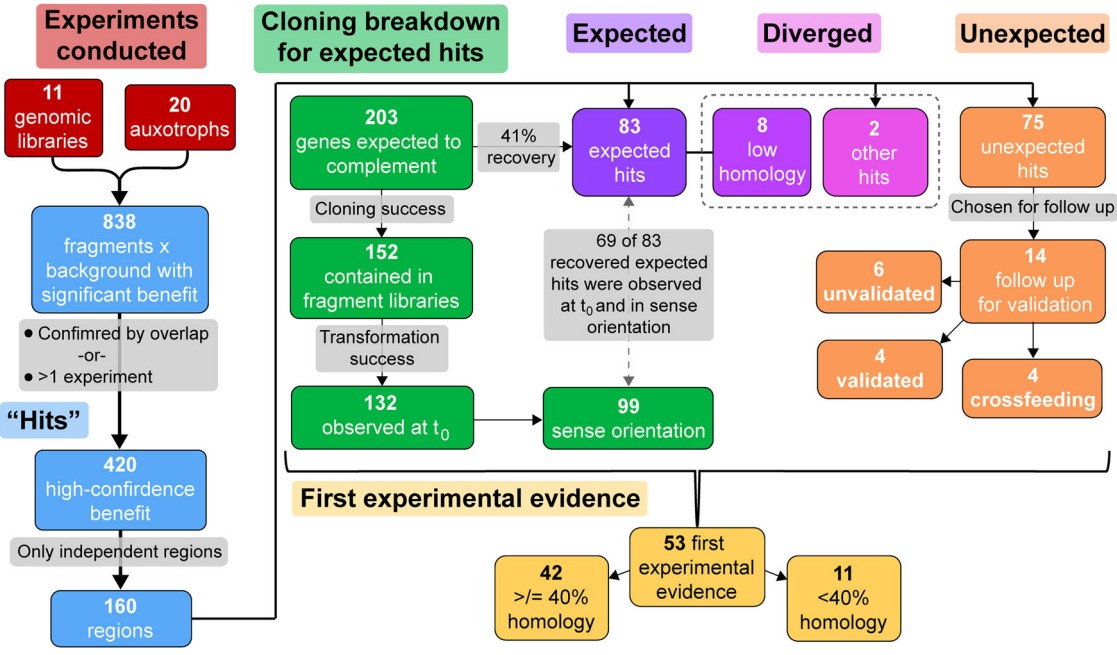

**Figure 9.  Overview of Coaux-Seq experiments and findings.**

Figures gives a flowchart for various findings of this work, including the experiments conducted, breakdown categories of findings, and results from controls.

providing a new expression vector and introducing diverse bacteria and auxotrophy as a testing context. This method compliments loss-of-function tools like RB-TnSeq (Price et al, 2018; Wetmore et al, 2015) and various CRISPRi strategies (Enright et al, 2024). Similar to how RB-TnSeq dramatically improved the throughput of transposon-based loss-of-function experiments that have been employed for many years (Hensel et al, 1995; Cain et al, 2020), Coaux-Seq significantly improves the throughput of gene complementation studies, which have been utilized for even longer (Fincham, 1968). These modern techniques owe their improved throughput over historical characterization approaches to advances in multiplexing and next-generation sequencing (NGS), as covered thoroughly by Gray and coworkers (Gray et al, 2015). Coaux-Seq

accesses this benefit by creating DNA-barcoded overexpression libraries from randomly sheared genomic fragments and then mapping with long-read sequencing technology (PacBio) to enable repeated assaying of contained genes to complete missing biochemical function. By utilizing BarSeq, which leverages primers containing Illumina adapters and only requires short sequencing read length, it also is cost-effective. Ultimately, Coaux-Seq not only provided the first experimental validation for the activity of 53 proteins, but also uncovered novel function. Figure 9 provides a flowchart style summary of the findings of this study including hit identification, hit categorical breakdown, expected hit recovery, and novel findings (see Supplemental Note in Appendix for additional discussion).

Having established this workflow, it could readily be extended and in multiple ways. First, additional genomic libraries could be generated for newly isolated species. Second, additional genetic knockout backgrounds (auxotrophies) could be tested in *E. coli*, or in different species that may have different expression preferences (e.g., alpha-proteobacteria), although this may require a different expression vector. Alternatively, expression hosts like *E. coli* could be modified to improve the expression of heterologous DNA such as through provision of rare tRNAs (Cheng et al, 2015). Additional auxotrophies utilized could include genetic contexts that extend beyond single gene knockouts to target desired biochemical activity. Schema besides auxotrophies could also be used. Third, this assay could be extended to other sources of DNA beyond that of isolated bacterial genomes, such as field-derived metagenomic DNA. However, this would require a more efficient assembly method for the barcoded libraries beyond blunt-end ligation, such as one based on tagmentation (Crofts et al, 2021), to improve library diversity. All of these future assays would be benefited by improved transformation efficiency and improved means of high-throughput, particularly liquid, growth assays that do not bias the libraries to a few top candidates. Importantly, we have highlighted several ways in which one might observe false positives and false negatives in our complementation assay workflow for gene identification, providing a template for future users of such a method. On this note, future studies may consider adjusting their fitness value cutoffs. If we had increased our fitness cutoff from 5 to 6, we would have observed a decrease in unexpected hits, which likely contain false positives (from 75 to 40), but also a slight decrease in recovered expected hits (from 83 to 80). The optimum cutoff may be context-dependent. Ultimately, Coaux-Seq provides a useful advance in functional genomics and has the potential to play an important role in future genetic annotation, particularly for core metabolism among non-model microbes. Improvements in such annotation stand to benefit downstream modeling approaches, including genome-scale metabolic models, with application in monoculture and microbial community contexts.

# Methods

### Reagents and tools table

| Reagent/resource | Reference or source | Identifier or catalog number |
|---|---|---|
| **Experimental models** | | |
| *Sphingomonas koreensis* JSS26 | DSM | 15582 |
| *Escherichia coli* BW25113 | Yale Coli Genetic Stock Center | 7636 |
| *Bacillus subtilis* 168 | Bacillus Genetic Stock Center (Ohio State University) | 1A1 |
| Bacteroides thetaiotaomicron VPI-5482 | ATCC | 29148 |
| *Pseudomonas fluorescens* FW300-N2E2 | Price et al (2018) | |
| *Rhodanobacter denitrificans* FW104-10B01 | Carlson et al (2019) | |
| *Lysobacter sp.* FW306-1B-D06B | This study | |

| Reagent/resource | Reference or source | Identifier or catalog number |
|---|---|---|
| *Pedobacter sp.* FW305-3-2-15-E-R2A2 | This study | |
| *Xylophilus sp.* GW821-FHT01B05 | This study | |
| *Rhodoferax sp.* GW822-FHT02A01 | This study | |
| *Acidovorax sp.* FHTAMBA | This study | |
| *Escherichia coli* ΔaroA | Baba et al (2006) | |
| *Escherichia coli* ΔthrB | Baba et al (2006) | |
| *Escherichia coli* ΔmetB | Baba et al (2006) | |
| *Escherichia coli* ΔmetC | Baba et al (2006) | |
| *Escherichia coli* ΔproB | Baba et al (2006) | |
| *Escherichia coli* ΔproA | Baba et al (2006) | |
| *Escherichia coli* ΔcysA | Baba et al (2006) | |
| *Escherichia coli* ΔargG | Baba et al (2006) | |
| *Escherichia coli* ΔleuA | Baba et al (2006) | |
| *Escherichia coli* ΔtrpA | Baba et al (2006) | |
| *Escherichia coli* ΔhisC | Baba et al (2006) | |
| *Escherichia coli* ΔmetE | Baba et al (2006) | |
| *Escherichia coli* ΔilvD | Baba et al (2006) | |
| *Escherichia coli* ΔpheA | Baba et al (2006) | |
| *Escherichia coli* ΔpurE | Baba et al (2006) | |
| *Escherichia coli* ΔcysH | Baba et al (2006) | |
| *Escherichia coli* ΔhisG | Baba et al (2006) | |
| *Escherichia coli* ΔaroE | Baba et al (2006) | |
| *Escherichia coli* ΔpyrD | Baba et al (2006) | |
| *Escherichia coli* Δppc | Baba et al (2006) | |
| **Recombinant DNA** | | |
| pBbA2c-RFP | Addgene | Cat. #35326 |
| pBWB507 | Addgene | Cat. #209325 |
| pBWB514 | Addgene | Cat. #209326 |
| **Oligonucleotides and other sequence-based reagents** | | |
| PCR Primers | This study | Dataset EV8 |
| **Chemicals, enzymes, and other reagents** | | |
| PrimeSTAR Max DNA Polymerase | Takara | R045A |
| Transposase EZ-TN5 | VWR (Lucigen) | 75927-976 (TNP92110) |
| Fast DNA End Repair Kit | Thermo Fisher | FERK0771 |
| SMRTbell Prep Kit 3.0 | PacBio | 102-182-700 |
| AMPure PB Beads | PacBio | 100-265-900 |
| Exo CIP Rapid PCR Cleanup Kit | New England Biolabs | E1050S/L |
| KLD Enzyme Mix | New England Biolabs | M0554S |
| T4 DNA Ligase (2,000,000 units/ml) | New England Biolabs | M0202M/T |
| T4 DNA Ligase Rxn Buffer | New England Biolabs | B0202S |

| Reagent/resource | Reference or source | Identifier or catalog number |
|---|---|---|
| T4 Polynucleotide Kinase | New England Biolabs | M0201S |
| dATP solution | New England Biolabs | N0440S |
| Taq polymerase with ThermoPol Buffer | New England Biolabs | M0267S |
| Shrimp Alkaline Phosphatase (rSAP) | New England Biolabs | M0371S |
| NEBuilder HiFi DNA Assembly Master Mix | New England Biolabs | E2621S |
| SapI (enzyme) | New England Biolabs | R0569S |
| 5-alpha competent *E. coli* | New England Biolabs | C2987H |
| 10-β high-efficiency chemically competent *E. coli* | New England Biolabs | C3019H |
| 10-β high-efficiency electrocompetent *E. coli* cells | New England Biolabs | C3020K |
| ATP | Fisher Scientific | FERR0441 |
| GeneJET Plasmid Miniprep Kit | Thermo Fisher | K0502/3 |
| GeneJET Gel Extraction Kit | Thermo Fisher | K0691 |
| GeneJET Genomic DNA Extraction Kit | Thermo Fisher | K0721 |
| GeneJET PCR Purification Kit | Thermo Fisher | K0701 |
| DpnI (enzyme) | Thermo Fisher | ER1701 |
| GeneRuler 1 kb Plus DNA Ladder, ready-to-use | Thermo Fisher | SM1333/4 |
| SybrSafe | Thermo Scientific (Invitrogen) | S33102 |
| Nuclease Free Water | Promega | P1197 |
| 50x TAE | Omega Bio-Tek | AC10089 |
| Agarose (UltraPure) | Thermo Scientific (Invitrogen) | 16500-500 |
| Anhydrotetracycline HCl | VWR (Adipogen) | 102989-258 (CDX-A0197-M500) |
| Dimethyl sulfoxide (DMSO) | Sigma-Aldrich | D8418-50ML |
| PCR tubes (0.5 mL) | Eppendorf | 30124537 |
| 1.5 ML tubes, sterile | VWR | 76332-072 |
| 14 ML TBE Snap Culture tubes | Fisher Scientific (Falcon) | 1495911B (352059) |
| 50 mL screwcap tubes | Fisher Scientific (Falcon) | 14-432-22 (352070) |
| Cryovials | VWR | 89125-508 |
| VWR FILTER UNIT PES 75MM 0.2U 500 ML CS12 | VWR | 10040-436 |
| Corning™ Disposable Vacuum Filter/Storage Systems | Fisher Scientific (Corning) | 09-761-102 (430758) |
| Rapid-Flow Sterile Disposable Filter Units (PES Membrane, 250 mL, 0.2 m pore, 50 mm membrane) | Nalgene | 568-0020 |
| Ethyl Alcohol, (Pure, 200 Proof) for molecular biology | Sigma-Aldrich | E7023-500ML |

| Reagent/resource | Reference or source | Identifier or catalog number |
|---|---|---|
| Electroporation cuvette | VWR | 89047-206 |
| miniTUBE Blue 3.0 kb | Covaris | 520065 |
| 10x Phosphate Buffered Saline, Sterile | Sigma-Aldrich | P5493 |
| LB Broth Miller | Fisher Scientific (BD Difco) | DF0446-07-5 |
| R2A Broth | HIMEDIA | M1687-500G |
| Agar | Fisher Scientific (BD Difco) | DF0812-07-1 |
| Glass Beads | Fisher Scientific (Biomxy) | NC9934837 |
| D-glucose (anhydrous) | VWR | 97061-166 |
| M9 Minimal Salts 5X | Fisher Scientific (BD Biosciences) | DF048517 |
| Glycerol | Sigma-Aldrich | G5516 |
| Magnesium sulfate | Sigma-Aldrich | M2643-500G |
| Calcium chloride | Sigma-Aldrich | C5670-100G |
| Kanamycin sulfate from Streptomyces kanamyceticus | Sigma-Aldrich | K1377-1G |
| Chloramphenicol | Sigma-Aldrich | C0378 |
| 10-β Stable Recovery Medium | New England Biolabs | B9035S |
| SOC Media | New England Biolabs | B9020S |
| **Software** | | |
| Illustrator | https://www.adobe.com/products/illustrator.html | |
| R (4.4.1) | https://www.r-project.org/ | |
| MultiCodes.pl | https://bitbucket.org/berkeleylab/feba | |
| Boba-seq | https://github.com/OGalOz/Boba-seq | |
| NCBI BLAST+ (2.13.0 + ) | https://blast.ncbi.nlm.nih.gov/doc/blast-help/downloadblastdata.html | |
| GapMind | http://papers.genomics.lbl.gov/gaps | |
| TIGRFAMs | https://tigrfams.jcvi.org/cgi-bin/index.cgi | |
| HMMR (3.3.1) | http://hmmer.org/ | |
| PaperBLAST | http://papers.genomics.lbl.gov/ | |
| EMBOSS (6.6.0) | http://emboss.open-bio.org/ | |
| OSTIR (1.1.0) from bioconda | | |
| **Other** | | |
| Innova 42R Shaker Incubator | Eppendorf | M1335-0014 |
| UV-VIS Spectrophotometer | Shimadzu | UV-1280 |
| NanoDrop 2000 | Thermo Scientific | ND-2000 |
| C1000 Thermal Cycler | Bio-RAD | 1851148 |
| S220 | Covaris | 500217 |

| Reagent/resource | Reference or source | Identifier or catalog number |
|---|---|---|
| Water Chiller MM7 | VWR | 13271-204 |
| Electroporator (PrecisionPulse) ECM 630 | BTX Harvard Apparatus | MA1 45-0051 |
| Gel Doc XR+ | Bio-RAD | 1708195EDU |
| Microcentrifuge (Refrigerated) | Eppendorf | 5417R |
| Microcentrifuge | Eppendorf | 5418, 05-401-202 |
| Centrifuge (Refrigerated) | Eppendorf | 5810R, 05-413-112 |
| Vortexer | VWR | 76549-928 |
| Minicentrifuge (MiniFuge) | VWR | 76269-066 |
| Power Source | VWR | 75848-722 |
| Mini-Sub Cell GT Horizontal Electrophoresis System | Bio-RAD | 1704467 |
| Freezer (−20 °C) | pHcbi | MDF-MU539HL-PA |
| Freezer (−80 °C) VIP plus+ | pHcbi | MDF-DU901VHA-PA |

## Culturing and media

Cultivations were incubated in a New Brunswick Innova 42 R incubator. General cultivations were carried out in LB Miller medium at 37 °C. For liquid cultures, 200 rpm shaking was used. After transformation, cells were recovered in either New England Biolabs (NEB) SOC or 10-β/stable recovery medium. Complementation selections were carried out in Difco M9 medium with calcium and magnesium added, and with 1% glucose as a carbon source. All agar plates were 1.5% agar (15 g agar per 1 L). Kanamycin, used for Escherichia coli Keio single gene knockout strain selection, was used at a concentration of 25 μg/mL. Kanamycin was prepared as 1000× stocks in water, sterile filtered (0.2 μm), and stored as 1 mL aliquots in sterile 1.5-mL Eppendorf tubes at −20 °C. Chloramphenicol, which was used for plasmid selection, was used at a concentration of 17 μg/mL. Chloramphenicol was prepared as 500× stocks in 95% ethanol, sterile filtered (0.2 μm), and stored as 1 mL aliquots in sterile 1.5-mL Eppendorf tubes at −20 °C. Anhydrotetracycline (aTc), used for synthetic promoter (tet) induction, was primarily used at 100 μg/mL ("1×") unless otherwise noted. aTc was first prepared as a 100,000× stock in DMSO, the diluted to 1000× stocks in DMSO as needed. aTc stocks, within a sterile 50-mL Falcon tube or a sterile 1.5-mL Eppendorf tube, were wrapped in aluminum foil to protect from light degradation and stored at −20 °C.

## Cloning strain, strain isolation, and genomic sequencing

Genome fragment library generation cloning was carried out in New England Biolabs 10-β high-efficiency electrocompetent E. coli cells. Individual knockout variants of Escherichia coli BW25113 for

complementation transformations were generous gifts of Dr. Gareth Butland and were originally obtained from the Keio collection (Baba et al, 2006). Sphingomonas koreensis JSS26 DSMZ 15582 was obtained from DSM. Escherichia coli BW25113 wild-type was obtained from the Coli Genetic Stock Center at Yale University. Bacillus subtills 168 was obtained from the Bacillus Genetic Stock Center at The Ohio State University. Bacteroides thetaiotaomicron VPI-5482 was obtained from ATCC. Pseudomonas fluorescens FW300-N2E2, Lysobacter sp. FW306-1B-D06B, Xylophilus sp. GW821-FHT01B05, Rhodanobacter denitrificans FW104-10B01, Rhodoferax sp. GW822-FHT02A01, Pedobacter sp. FW305-3-2-15-E-R2A2, and Acidovorax sp. FHTAMBA were isolated from the Oak Ridge Reservation Field Site.

The full description of the isolation for P. Fluorescens FW300-N2E2 has been given previously (Price et al, 2018). The isolation for R. denitrificans FW104-10B01 has been described elsewhere as well (Carlson et al, 2019). Lysobacter sp. FW306-1B-D06B was isolated at 25 °C, aerobically, in R2A medium in 96-well carbon from sample FWB306-02. Pedobacter sp. FW305-3-2-15-E-R2A2 was isolated aerobically in R2A medium at 30 °C from sample FW305-03-02-15. Xylophilus sp. GW821-FHT01B05 was isolated aerobically in R2A medium at 25 °C from sample GW821-2019-05-13-UF-R10. Rhodoferax sp. GW822-FHT02A01 was isolated using ground water pre-incubated at 4 °C aerobically in R2A medium at 25 °C from the GW822E-2019-04-01-UF-R10 sample as part of a high-throughput campaign. Acidovorax sp. FHTAMBA was isolated as part of high-throughput isolations from the Oak Ridge site. Previously sequenced genomes were available prior to this study for E. coli BW25113 (Grenier et al, 2014), B. thetaiotaomicron VPI-5482, B. subtilis 168, P. fluorescens FW300-N2E2 (Carim et al, 2021), S. koreensis JSS26, and R. denitrificans FW104-10B01 (Peng et al, 2022). For all other strains, the genomes were sequenced. For Acidovorax sp. FHTAMBA, Xylophilus sp. GW821-FHT01B05, Lysobacter sp. FW306-1B-D06B, and Pedobacter sp. FW305-3-2-15-E-R2A2, Plasmidsaurus (Oregon, USA) hybrid Nanopore/Illumina sequencing and assembly services were utilized. For Rhodoferax sp. GW822-FHT02A01 HMW DNA was extracted using the Genomic Tip 100/G kit. Nanopore and Illumina libraries were prepped in-house and sequenced as described in Goff et al, 2022 (Goff et al, 2022). Sequence quality control is also the same as in Goff et al, 2022. The genome was hybrid assembled using Unicycler v0.4.8 (Wick et al, 2017) with default parameters. Each newly sequenced genome has been deposited to NCBI (accession number for newly sequenced genomes are also in the "Data availability" section). The full set of accession number for genomes used in this study include, CP009273.1/GCF_000750555.1 (Escherichia coli BW25113), NZ_CP015225.1/GCF_001623525.1 (Pseudomonas fluorescens FW300-N2E2), GCF_002797435.1 (Sphingomonas koreensis JSS26; DSMZ 15582), NC_000964.3/GCF_000009045.1 (Bacillus subtilis 168), NC_004663.1/GCF_000011065.1 (Bacteroides thetaiotaomicron VPI-5482), CP088922.1/NZ_CP088922.1/GCF_021560695.1 (Rhodanobacter denitrificans FW104-10B01), CP151802 (Lysobacter sp. FW306-1B-D06B), CP151803 (Pedobacter sp. FW305-3-2-15-E-R2A2), CP152408 (Xylophilus sp. GW821-FHT01B05), CP152407 (Acidovorax sp. FHTAMBA) and CP152376 (Rhodoferax sp. GW822-FHT02A01).

## General cloning

The plasmid backbone pBbA2c-RFP (Addgene #35326) (Lee et al, 2011), a generous gift from Prof. Jay Keasling, was modified to generate

the plasmids in this study. The updated plasmid contains a p15A origin of replication, an oriT to enable conjugation, a Tet promoter with a TetR repressor, a chloramphenicol resistance marker, and type II restriction sites (SapI). A plasmid map has been provided with the supplemental materials. The initial three genome fragment libraries (*E. coli* BW25113, *Pseudomonas fluorescens* FW300-N2E2, *and Sphingomonas koreensis*) were created in the context of the pBBa2c-RFP derivative pBWB507, a variant of which containing a ribosomal binding site and mRFP in place of a gene fragment has been deposited to Addgene (Addgene #209325). A slight modification to this plasmid was made to include SapI type II restriction sites, pBWB514 (Addgene #209326). The additional restriction sites make possible the transfer of mapped libraries, meaning those with long-read sequencing to connect the gene fragments to unique barcodes, to new vector backgrounds without the need for PCR, which could induce library bias. All of the other eight libraries were cloned into pBWB514. Both vectors contain mosaic ends (Crofts et al, 2021) to potentially allow for Gibson assembly cloning in future genome fragment library creation. Plasmid modifications were generated by successive rounds of PCR using PrimeSTAR by Takara following manufacturer protocols, (3 min 98 °C denaturing, followed by 30× cycles of at a 10 s 98 °C denaturing step, a 15 s 55 °C annealing step, and a 45 s 72 °C extension step, followed lastly by a 5 min 72 °C extension). Self-ligation was used to create the new plasmids using NEB's KLD kit, following manufacturer protocols. All primers used for plasmid modification, sequencing, and analysis are provided as an Excel sheet as Dataset EV7. GenBank files for newly developed plasmids are provided with Dataset EV8.

## Genomic fragment library preparation

Libraries were prepared by first growing the individual strains to an $OD_{600} > 1.0$ (37 °C for *E. coli*; 30 °C for all other strains). Cultures were carried out at 5 mL volume in 14-mL Falcon culture tubes. Genome extractions were carried out with a Thermo Scientific GeneJET Genomic DNA Purification kit, following the manufacturer's protocols, eluting into nuclease-free water, and adjusting for gram-negative or gram-positive bacteria. Following extraction, genomes were sheared to a mean 3-kb fragment size with a Covaris S220. Between 2 and 20 µg of genomic DNA (typically 2 µg, total final volume 200 µL) was added into a Covaris blue miniTUBE, and the equipment was run using the 3-kB setting (SonoLab 7.2 software, temperature set between 4 and 25 °C, Peak Power 3.0, Duty Factor 20.0, Cycles/Burst 1000, 600 s). Following sonication, sheared genomes were run on a 1% agarose gel (100 volts, 400 mA, ~25 min), and the "smeared" band between 1.5 and 5 kb was excised (Appendix Fig. S12 for an example). Sheared genomic DNA was collected via gel extraction using a Thermo Scientific GeneJET Gel Purification kit, following the manufacturer's instructions but eluting into nuclease-free water.

After gel extraction, genomic fragments were end-repaired using a Thermo Scientific Fast DNA End Repair Kit (manufacturer's protocols). Between 2 and 4×50 µL reactions were run and subsequently pooled using a Thermo Scientific GeneJET PCR Purification Kit (manufacturer's protocols, nuclease-free water elution). Even though the Fast DNA End Repair Kit phosphorylates the end-repaired DNA fragments, for thoroughness, additional phosphorylation was conducted with NEB T4 Polynucleotide Kinase (PNK), following the manufacturer's protocols and using the 1× T4 DNA Ligase Buffer that contains 1 mM ATP. Multiple 50 µL reactions were

run to include all end-repaired DNA. After phosphorylation, all reactions were again pooled using the Thermo Scientific GeneJET PCR purification Kit and eluted with 20 µL of nuclease-free water.

The backbone was prepared by PCR to introduce barcode diversity via primer overhang. For pBWB507, to linearize and add the barcode and flanking regions used for BarSeq (described below), the forward primer used was BWB1429 (GTCGACCTG CAGCGTACGNNNNNNNNNNNNNNNNNNNNNAGAGACCTC GTGGACATCactgtcTAGGGATAACAGGG) and the reverse primer was BWB1422 (atggtctgaattctttctctatcac). Similarly, for pBWB514, BWB1471 (CTGTCTCTTATACACATCTGTCGACCT GCAGCGTACGNNNNNNNNNNNNNNNNNNNNNAGAGACCT CGTGGACATCG) and BWB1472 (CTGTCTCTTATACACATC TTGGACTGAAGAGCtttctctatc) were used. All primers were ordered from IDT. None of these primers were phosphorylated and BWB1429 and BWB1471 were HPLC-purified to help ensure full-length products from PCR. Backbone preparation was run in 4× of 50 µL PCR using PrimeSTAR Max from Takara as described above. Following PCR, 1 µL of DpnI was added to each PCR reaction and the reaction was incubated at 37 °C for 1 h to remove the PCR template plasmid. Following, a 1% agarose gel was run, and the ~3 kb backbone was excised and purified by gel extraction. Out of an abundance of caution, an NEB rSAP reaction was run on the purified backbone to ensure no phosphorylation of the backbone to avoid self-ligation during blunt-end ligation cloning. After the rSAP reaction, another GeneJET PCR purification was run.

### Blunt-end ligation, genome fragment library transformation

Once both insert and backbone DNA elements were prepared as described above, blunt-end ligation with NEB T4 DNA ligase was conducted (2,000,000 units/mL). The ligations were carried out at a 1:3 backbone to insert molar ratio, run in 4× at a 20 µL volume, and reactions were run at 16 °C overnight (~16 h). After overnight incubation, the 4× ligations were pooled via a GeneJET PCR Cleanup Kit and the DNA concentration was measured with a Thermo Scientific NanoDrop 2000 using the NanoDrop 2000 software. Following, 3–5 µL of this pooled ligation product was added to NEB 10-β high-efficiency electrocompetent cells (25 µL) in a VWR 90 µL sterile electroporation cuvette and were electroporated with a BTX Harvard Apparatus Electro Cell Manipulator PrecisionPulse (ECM 630, Version 1.05) with a BTX Harvard Apparatus Safety Stand (630B). Electroporation settings were 1750 volts, 200 Ω resistance, and 25 µF capacitance. Immediately after electroporation, cells were recovered with 975 µL of 10-β/stable NEB recovery medium for 1 h at 37 °C and subsequently stored at 4 °C until later use. To ensure sufficient library size, up to 4 separate ligation product transformations were conducted per library.

### Plating, library outgrowth

To determine transformation efficiency, tenfold serial dilutions in sterile 1× PBS were plated as 15-µL spots on LB agar chloramphenicol plates. After drying, plates were incubated at 37 °C overnight. The next morning colonies were counted to determine the colony-forming units per volume (cfu/mL), and thus determine the volume of recovered electroporation media necessary to target a generated a fragment library to sufficiently "cover" or sample the genome. Coverage need was estimated by taking the number of putative genes in the genome (typically 3000–6000) and

multiplying by ~10× (30,000–60,000 cfus, typically). This calculated volume of the recovered electroporation was then added to 50 mL of LB medium with chloramphenicol and allowed to grow overnight in a 250 mL Erlenmeyer flask. The next morning, 4× glycerol stocks (1 mL volume equal parts 50% glycerol and culture medium) were prepared and 6 × 5 mL aliquots of the culture medium were used for a miniprep to recover the plasmid library using a Thermo Scientific GeneJET Plasmid Miniprep Kit, following manufacturer's instruction but eluting with nuclease-free water. Cultures aliquoted for the minipreps were centrifuged for 10 min at 4000 × g at 4 °C to pellet the cells. After decanting the media, the remaining media was gently removed by pipetting. The plasmid library obtained from miniprep was subsequently used for BarSeq to determine library size by sequencing, PacBio sequencing to further validate library size and quality and to link gene fragment to barcodes, and complementation transformations.

## PacBio sequencing

Preparation for long-read sequencing began with $4 \times 50\,\mu L$ PCR, using PrimeSTAR Max, as described above with the exception that 100 ng of plasmid library was used as template and with only 10× PCR cycles. For this PCR, both primers were HPLC-purified and 5′ phosphorylated. For libraries on pBWB507 the primers were (Fw-/5Phos/gtgatagagaaaagaattcagaccat and Rv-/5Phos/cagtGATGTCCACGAGGTCTC) and for pBWB514 (Fw-/5Phos/GTCCAAGATGTGTATAAGAGACAG and Rv-/5Phos/GACGATGTCCACGAGGTCTC). After PCR, the product was run on 1% agarose gel and excised to gather the 1–5 kb range and pooled. After gel extraction, the PCR product was digested with DpnI at 37 °C for 1 h. After digestion, reactions were pooled using a GeneJET PCR purification. From this point, the PCR products of the DNA fragment libraries were prepared for PacBio sequencing using the PacBio SMRTbell prep kit 3.0, following the adapter-barcoded protocol (Barcoded overhang adapter kits 8A and 8B), including the initial bead cleanup. Barcoded libraries were pooled and submitted for sequencing with the University of California-Berkeley QB3 Genomics core and sequenced with their PacBio Sequel II (SMRT cell 8 M).

### Computational pipeline used to map barcodes to genes

The Boba-seq script was used to map barcodes to genomic inserts and to genes and can be found at https://github.com/OGalOz/Boba-seq (Huang et al, 2022). Briefly, assembly sequence files (FASTA) and annotation files (GFF/GFF3) for each source isolate were either downloaded from RefSeq or generated as part of sequencing work in this study. Default parameters were used for all steps as listed on the configuration. JSON file. PacBio's lima tool, usearch (www.drive5.com/usearch/), vsearch (https://github.com/torognes/vsearch), and minimap2 (Li, 2018) are used as part of the pipeline. First, PacBio CCS reads are demultiplexed and then the insert and barcode sequences are extracted. Reads with concatemers or with incorrect barcode lengths are filtered out. Only inserts with 10 or fewer expected errors are kept. Inserts are then mapped to the genome assembly, and a series of criteria are used to identify high-confidence mappings. Finally, genomic positions are mapped to protein-coding genes to generate final mapping tables used to calculate library statistics and to identify gene hits from complementation assays.

### Library generation and mapping protocol

A version of the protocol is maintained at protocols.io https://doi.org/10.17504/protocols.io.q26g71311gwz/v1.

1. Streak out glycerol stock of desired strain on appropriate agar plate (typically LB or R2A, 1.5% agar). Grow at appropriate temperature (often 30 °C or 37 °C) and time (usually overnight) to obtain colonies.

2. If colonies have formed, pick a colony and inoculate the strain into an appropriate liquid medium and scale volume of inoculation for necessary optical density ($OD_{600}$) equivalent to satisfy following genome extraction (typically 15–50 mL should suffice). Grow strains overnight at appropriate temperature conditions.

3. Measure $OD_{600}$ the next morning to perform cell count calculation/approximation.
   a. NOTE: When working with non-model microbes (such as those isolated from the environment), their $OD_{600}$ to cfu correlation is not inherently equal to model systems like *Escherichia coli*. Therefore, it can be valuable to separately run a dilution curve of $OD_{600}$ and plate the cultures to perform cell counts to determine this relationship.

4. If the desired cell count has been achieved, pellet an appropriate volume of cell for subsequent steps. Centrifugation conditions—4000 rpm, 5 min, 4 °C.

5. Use the preferred Genomic DNA extraction kit, following the appropriate manufacturer instructions for Gram-negative or Gram-positive bacteria. Here, we used the Thermo Scientific GeneJET Genomic DNA Extraction Kit (Thermo Scientific, K0721).
   a. NOTE: For subsequent steps, it is preferred to elute into nuclease-free water at the end of the manufacturer protocol instead of the kit provided elution buffer.

6. Measure DNA concentration (Nanodrop or Qubit).
   a. NOTE: If your DNA concentration is low, you may need to run multiple extractions, pool, and concentrate to achieve the required $2\,\mu g$ of genomic DNA in a <200 µL volume of nuclease-free water for subsequent steps. This is why it is better to elute in nuclease-free water than elution buffer, so as not to concentrate the salts when evaporating. Alternatively, one can do an additional DNA cleanup step and load multiple cleanup reactions onto a single column and elute into a smaller volume of nuclease-free water. If this option is chosen, be sure the column and protocol is suitable for high molecular weight DNA.

7. Take 2–20 µg (2 µg is frequently used) of genomic DNA and combine with nuclease-free water to a volume of 200 µL in a Covaris Blue miniTube (3 kb).

8. Turn on Covaris 220 and chiller (add water to sonication components). Ensure that machine gets to appropriate temperature and all air is cleared from system (all system checks are passed).

9. Add Blue miniTube containing genomic DNA to Covaris 220 using miniTube loading apparatus. Run 3 kb sonication protocol. (Covaris settings—SonoLab 7.2 software, temperature 4–25 °C, Peak Power 3.0, Duty Factor 20.0, Cycles/Burst 1000, 600 s).
   a. NOTE: Make sure there are no air bubbles under or around the tube before starting the protocol!

10. Once sonication protocol is complete, remove Blue miniTube containing sheared genomic DNA (turn off machine, drain water thoroughly from sonication components) and utilize

sheared genomic DNA for subsequent steps.

11. Add appropriate volume of DNA gel loading dye to Blue miniTube.
    a. NOTE: Most loading dyes, including the one provided with the GeneRuler 1 kb+ kit (Thermo Scientific, SM1333/4), are 6× and would suggest adding 40 μL of dye to the sheared DNA. However, using less loading dye (~20 μL) will be sufficient and prevent dilution of fragmented genomic DNA.

12. Cast a 1% agarose gel (in 1× TAE buffer) with appropriate amount of SybrSafe (Invitrogen, S33102) or equivalent DNA imaging reagent. Here, we use 50x TAE buffer and dilute (Omega, AC10089) and UltraPure Agarose (Invitrogen, 16500-500).

13. Once gel is set, load ladder (Generuler 1 kb + ) and genomic fragment material with loading dye added.

14. Run gel at 100 v, 400 mA, for 25–30 min.

15. Image gel with appropriate gel imaging equipment (e.g., Bio-RAD Gel Doc XR + ). One should observe a smear of DNA with the strongest brightness centered at 3 kb.

16. Cut desired segment of DNA from the gel (recommend ~1–5 kb section).

17. Store cut bands in 1.5–2 mL Eppendorf tubes. Use gel slices for the following step.

18. Use GeneJET Gel Extraction Kit (Thermo Scientific, K0691) or preferred gel extraction kit to extract genomic fragment DNA.
    a. NOTE: When in doubt, use greater volumes of binding buffer to dissolve gel. In addition, add binding buffer to column and do an extra spin before loading genomic fragment DNA (helps retention as column is better equilibrated). Perform an extra loading buffer wash before ethanol-based wash steps. Elute into desired volume of nuclease-free water (30–50 μL). Measure DNA concentration (NanoDrop or Qubit).

19. Take gel extracted DNA and repairs ends and phosphorylate using Fast DNA End Repair Kit following manufacturer protocols (Thermo Fisher Scientific, FERK0771).

20. As a precaution, although the end repair kit should phosphorylate the ends of your fragmented genomic DNA, run one additional phosphorylation step, using manufacturer instructions with T4 Polynucleotide Kinase (PNK) kit (New England Biolabs, M0201S).

21. After phosphorylation, run a PCR cleanup using GeneJET PCR Purification Kit (Thermo Scientific, K0701) or preferred alternative to prepare for subsequent ligation step. Measure DNA concentration (NanoDrop, Qubit). **The genomic fragment library is now prepared for cloning into expression vector**.

22. To generate the vector backbone for library cloning, use plasmid pBWB514 (Addgene #209326) as a PCR template, with: Forward primer BWB1471 (CTGTCTCTTATACACATCTGTCGACCT GCAGCGTACGNNNNNNNNNNNNNNNNNNNNNNNAGAGAC CTCGTGGACATCG), Reverse primer BWB1472 (CTGTCTC TTATACACATCTTGGACTGAAGAGCttttctctatc).
    a. Use PrimeSTAR Max 2× master mix (Takara, R045A) following manufacturer's protocols. This PCR will both linearize the plasmid for blunt-end cloning (conducted later) and add the barcode and other desired genetic parts (mosaic ends, U1, U2).
    b. NOTE: Primers should not be phosphorylated (to prevent self-ligation in subsequent steps) and should be purified

(such as by HPLC) when ordering to ensure full-length primers are used during PCR.
    c. NOTE: Use a low concentration of template plasmid DNA (~5 ng) to prevent false positives at the later transformation step (from DNA carryover).
    d. NOTE: To prepare sufficient backbone for the creation of multiple genome libraries, run 4 × 50 μL PCR reactions.

23. As a precaution, to remove the remaining template plasmid DNA, run a dpnI digest with the PCR reaction.
    a. Simple approach: Add 1 μL of dpnI (Thermo Scientific, ER1701) to 50 μL PCR reaction and run at 37 °C for 1 h.
    b. Alternatively, one can run a PCR cleanup such as with a GeneJET PCR Purification Kit (Thermo Scientific, K0701), and following run a dpnI digest according to the manufacturer protocols.

24. After the PCR and dpnI digest is complete, run a 1% agarose gel as described previously.

25. Gel extract the linearized plasmid band with a GeneJET Gel Extraction Kit (Thermo Scientific, K0691). The desired band should be 2976 bp (~3 kb). Elute and measure DNA concentration (NanoDrop or Qubit).

26. As a precaution, run rSAP (Shrimp alkaline phosphatase) dephosphorylation reaction on backbone according to the manufacturer protocols (New England Biolabs, M0371S).

27. Clean up DNA with GeneJET PCR Purification Kit (Thermo Scientific, K0701). Measure DNA concentration (NanoDrop or Qubit). **The backbone is now prepared for fragment library cloning**.

28. Using a high concentration T4 DNA ligase (New England Biolabs, M0202M), run a blunt-end ligation reaction following the manufacturer's protocols. Utilize 2:1 or 3:1 insert to backbone molar ratio. As the backbone (2976 bp) and mean insert size (3 kb) are approximately the same, one can use a mass ratio for this step. When running ligation, utilize the 16 °C overnight option.
    a. NOTE: Too high of insert amount will lead to a higher frequency of the insert ligating to insert. Too low of insert amount will lead to fewer successful ligation events.

29. Because of the desired library size, run 4 parallel ligations (20 μL each) and pool using a single PCR cleanup step with a GeneJET PCR Purification Kit (Thermo Scientific, K0701). Elute into 18 μL of nuclease-free water to prepare for electroporation. If desired, measure DNA concentration and quality (NanoDrop or Qubit).

30. Use ~3 μL of pooled and purified ligation product to transform 10-β high-efficiency electrocompetent *E. coli* cells following manufacturer protocols (New England Biolabs, C3020K). It is recommended to run 3 or 4 transformations for each library.

31. After the manufacturer specified recovery of transformed cells, create a dilution series in sterile 1× PBS and spot on LB +chloramphenicol plates (17 μg/mL chloramphenicol concentration) to determine transformation efficiency. Store recovered cells at 4 °C overnight in recovery medium. Incubate dried, spotted plates at 37 °C.

32. The following morning, check the spotting plates to get colony-forming unit (cfu) counts to determine transformation efficiencies for each transformation run. Based on the desired library size (calculated by total genome size, divided by 300—3 kb fragments, 10× coverage), combine volumes of

the individual transformations to get to total desired cfu amount.

 a. NOTE: cfus often overestimate library size. Thus, to be conservative, it is best to have slightly more volume than calculated.

33. Transfer desired combination of individual transformation to 100 mL of liquid LB + chloramphenicol (17 μg/mL) in a non-baffled 500-mL shake flask. Grow for 8–10 h.

34. After outgrowth, make multiple (~10) glycerol stocks mixing cell culture and sterile 50% glycerol 1:1 (500 μL each) to make 1 mL stocks. Store glycerol stocks in appropriately labeled cryovials at −80 °C.

35. Plasmid miniprep the entirety of the remaining culture with a GeneJET Plasmid Miniprep Kit (Thermo Scientific, K0502).

36. Take cloned, plasmid miniprepped genomic fragment library and use as a template to prepare for PacBio sequencing. Following manufacturer protocols, use SMRTbell prep kit 3.0 to prepare library for PacBio Sequencing.

37. For PCR initial step, use HPLC-purified primers (Forward primer BWB1473-/5Phos/GTCCAAGATGTGTATAAGAGA-CAG and Rv primer BWB1474-/5Phos/GACGATGTCCAC-GAGGTCTC) and run a 10× cycle PCR with 100 ng template DNA using PrimeSTAR Max.

 a. NOTE: Run $4 \times 50$ μL PCR reactions to obtain sufficient material.

38. After PCR, run agarose gel as described previously and extract 1–5 kb band from gel with kit.

39. Run an additional dpnI digestion reaction on extracted DNA (37 °C, 1 h). Clean up DNA with a GeneJET PCR purification kit.

40. Follow manufacturer protocol for SMRTbell prep kit 3.0, including rigorous cleanup steps and DNA quantification.

41. Send for PacBio sequencing. Barcode and pool as needed with SMRTbell prep kit components.

42. Map libraries with https://github.com/OGalOz/Boba-seq. **Mapped libraries can now be used in Coaux-Seq complementation assays**.

### Preparation of Keio competent cells and electroporation

To prepare individual single gene knockout auxotrophic *E. coli* strains from the Keio collection for electroporation, they were first inoculated into 3 mL of LB with kanamycin and incubated overnight at 37 °C, 200 rpm shaking. The following morning, the overnight culture was diluted 1:100 into 50 mL of fresh medium and grown at 37 °C, 200 rpm shaking. The $OD_{600}$ was measured until the cells reached an $OD_{600}$ between 0.4 and 0.6. Following, the volume was split in half (~25 mL), and centrifuged for 10 min at $4000 \times g$ at 4 °C to pellet the cells. All the following steps were conducted on ice. After centrifugation, media was decanted and the cell pellet was resuspended in 5 mL of 10% glycerol that had been stored at 4 °C and kept on ice. Cells were pipetted gently to resuspend fully. After resuspension, cells were spun down again at $4000 \times g$ at 4 °C for 10 min. This 10% glycerol washing process was repeated 3 additional times. After the 4th wash, the cell pellet was resuspended into 100th of the original volume in 10% glycerol and stored at 30 μL aliquots in sterile 1.5 mL Eppendorf tubes and either stored at −80 °C or used immediately for transformation.

Electroporation was accomplished by adding 75 ng of plasmid DNA (3–5 μL volume) to the 30 μL competent cell aliquot, mixing by gently flicking the tube, transferring to an electroporation cuvette, and electroporating. After electroporating with a BTX Electro Cell Manipulator (PrecisionPulse) at 1750 V, 200 ohms resistance, and 25 μF capacitance, 975 μL of recovery medium was immediately added. For library recovery, LB medium was used. For selective recovery, complete M9 medium with 1% glucose was used. In either case, electroporated cells in the recovery medium were transferred to a 37 °C shaker incubator for 1 h. After 1 h, the recovered cells were either made into a glycerol stock (mixed 1:1 with 30% glycerol to make a 15% glycerol stock solution) and frozen at −80 °C, washed 3× in PBS before transfer to a selective condition (if in LB), or directly plated into a selective condition (if recovered in M9 medium).

### Complementation and selection of assay protocol

A version of the protocol is maintained at protocols.io https://doi.org/10.17504/protocols.io.j8nlk8m3wl5r/v1.

1. After determining the biochemical activity for which you want to screen, choose an appropriate auxotrophic strain and medium condition combination. For the following example, we will be using an *Escherichia coli* Keio single gene knockout strain and M9 minimal medium with 1% glucose as the sole carbon source.

 a. NOTE: Be sure to validate the knockout strain before use.

2. Streak out selected strain (from glycerol stock or other source) on appropriate medium (here LB 1.5% agar + 25 μg/mL kanamycin). Incubate at 37 °C overnight.

3. The following morning, pick a colony and inoculate into 3 mL of LB + 25 μg/mL kanamycin. Incubate at 37 °C, 150 rpm overnight.

4. (The next steps are typical preparation of electrocompetent *E. coli* and will be assuming the use of the Keio knockout strain.) The following morning, subculture 1:100 into 50 mL of fresh LB + 25 μg/mL kanamycin using a non-baffled 250 mL shake flask. Grow until an $OD_{600}$ of 0.4–0.6 is reached, measuring $OD_{600}$ periodically.

 a. NOTE: One can scale this depending on the number of aliquots of electrocompetent cells you would like to prepare.

5. Once desired $OD_{600}$ is obtained, pellet cells by centrifugation (4000 rpm, 5 min, 4 °C).

 a. NOTE: Run subsequent competent cell preparation steps on ice.

6. Wash cells 4-5x with sterile, cold (4 °C) 10% glycerol. Wash with a volume of at least 5 mL of 10% glycerol each time. For each step, gently resuspend by pipetting in the fresh, sterile, cold 10% glycerol. After, centrifuge cells (4000 rpm, 5 min, 4 °C). Repeat the process until the cells have been washed 4–5×.

 a. NOTE: Keep materials on ice, pay close attention to sterile technique. This step is to remove salts and other medium components that would causes issues during electroporation.

7. After the final wash, gently resuspend cells in 1:100 of original volume in 10% glycerol. (If you had a 50 mL culture, resuspend in ~500 μL of 10% glycerol).

8. In sterile 1.5-mL Eppendorf tubes, pre-chilled on ice, aliquot 30 μL of resuspended cells. **These cells are now prepared for electroporation**. They can either be used now (recommended for best transformation efficiency) or flash frozen and stored at −80 °C for later use.

9. Add 1–10 μL of highly purified cloned genomic fragment plasmid library (single genome library or combination) to 30 μL aliquot of electrocompetent cells. Flick tube gently 4–5 times to mix. Let sit on ice for 5 min.

a. NOTE: Continue to conduct steps on ice. The higher the purity of the transforming DNA, the greater volume you can add without issues during electroporation. If high efficiency is needed, one can perform an ethanol precipitation on the DNA and concentration highly before addition to competent cell mixture.

10. Transfer entire contents (mixed cells + plasmid library) to chilled (4 °C, on ice) electroporation cuvette.

11. Electroporate. Here we used a BTX Electro Cell Manipulator (PrecisionPulse) at setting 1750 V, 200 ohms resistance, and 25 µF capacitance.

12. If electroporation did not "arch," immediately added 975 µL of SOC recovery medium (NEB) and recover (37 °C, 1 h, 150 rpm). If electroporation "arched" discard cells.

13. After initial recovery, transfer to 100 mL of LB medium + 25 µg/mL kanamycin + 17 µg/mL chloramphenicol in a 500 mL non-baffled shake flask. Grow for 4–8 h (37 °C, 150 rpm), until culture becomes turbid.

    a. NOTE: Kanamycin is to select for Keio knockout. Chloramphenicol is to select for the plasmid.

14. Make multiple (4–10) glycerol stocks of rich medium recovered library to use for future selections. Use 1:1 culture and 50% glycerol (500 µL each) for a 1 mL final volume. Store in cryovials at −80 °C. Plasmid miniprep ~10 mL of culture for BarSeq to determine initial state of library ($t_0$) prior to selection. Use GeneJET Plasmid Miniprep Kit (Thermo Scientific, K0502) or equivalent for miniprep. Store $t_0$ miniprepped plasmid DNA in nuclease-free water at −20 °C until needed for BarSeq.

15. Take a glycerol stock of rich medium recovered library and inoculate entirety into 50 mL of LB + 25 µg/mL kanamycin + 17 µg/mL chloramphenicol in a 250 mL non-baffled shake flask. Incubate for 4–8 h (37 °C, 150 rpm), until culture becomes turbid.

16. Take culture, centrifuge to pellet (4000 rpm, 5 min, 4 °C). Wash 3× with sterile 1× PBS. For each wash, resuspend in 5 mL of sterile 1× PBS and centrifuge (4000 rpm, 5 min, 4 °C).

17. After washing steps, resuspend in 2 mL of sterile 1× PBS. Add 330 µL of resuspended cells to selection plates (here M9 minimal medium, 1.5% agar, 1% glucose with either 1× [100 ng/mL] or 5× [500 ng/mL] adenotetracycline [aTc] + 25 µg/mL kanamycin + 17 µg/mL chloramphenicol). Plate out one replicate per condition (aTc concentration). Use glass bead to spread evenly. Allow plates to dry in sterile environment (e.g., Biosafety cabinet).

    a. NOTE: aTc is used to induce the synthetic promoter.

18. Wrap plates with parafilm to prevent dehydration. Incubate plates at 37 °C until colonies are observed (this can take up to 1 week time).

19. Optional: Pick a few (~4) colonies to grow up in 5 mL M9 minimal medium, 1% glucose, 5× aTc, 25 µg/mL kanamycin + 17 µg/mL chloramphenicol (overnight) to miniprep and send for Sanger sequencing to "spot check" and confirm complementation success (meaning, do the genes identified make sense).

20. Once colonies are observed, use a colony scraper and scape the colonies into 5 mL of sterile 1× PBS. Pooled plates as desired. Once pooled, centrifuge to pellet the cells (4000 rpm, 5 min, 4 °C). Decant PBS.

21. Plasmid miniprep pooled pelleted cells using GeneJET Plasmid Miniprep Kit (Thermo Scientific, K0502) or equivalent. Store miniprepped plasmid in nuclease-free water at −20 °C until

preparation for BarSeq.

22. Run BarSeq on desired samples. Use protocol as described by Huang et al, 2022. Utilize https://github.com/morgannprice/BobaseqFitness.

### BarSeq

Barcodes were amplified using "BarSeq_V4" primers, as described previously (Huang et al, 2022). These use a 10-bp index sequence for P7 primers and an internal 8-bp index (to detect index hopping) and allow multiplexing up to 768 samples. The P7 BarSeq primers support demultiplexing by Illumina software via index reads, while the additional index added by the P5 primers is checked by the MultiCodes.pl script. BarSeq reads were converted to counts per barcode using the MultiCodes.pl script in the feba code base (https://bitbucket.org/berkeleylab/feba/), with the `-minQuality 0` option and `-bs4` (BarSeq 4) for BarSeq primers. To estimate the diversity of barcoded libraries, only reads that had a quality score of ≥30 at each position (the `-minQuality 30` option) were used, which corresponds to an error rate for barcodes of at most 0.001 * 20 nt = 2%. Furthermore, any barcodes that were off-by-1 errors from a more common barcode were eliminated. Per-strain fitness and z-scores were computed using bobaseq.R (https://github.com/morgannprice/BobaseqFitness) (Huang et al, 2022). These R scripts were also used to identify hits that were confirmed by overlap.

### Rare codon analysis

We computed the codon adaptation index (CAI) for each gene using the CAI program from EMBOSS (Rice et al, 2000) 6.6.0 with their reference table of codon usage in highly expressed genes of *E. coli*. We computed the frequency of codons that are rare in *E. coli* (ATA, CGG, CGA, CTA, AGA, AGG, GGA, or CCC) in each gene that was expected to complement one of the Keio knockouts. There was no significant difference in the total frequency of rare codons between expected hits and those that were not hits.

# Data availability

Source data for this study can be found on FigShare (https://doi.org/10.6084/m9.figshare.25749549). Links to GitHub repositories have been provided for code where relevant. Accession numbers for newly sequenced genomes include CP151802 (*Lysobacter sp.* FW306-1B-D06B—https://www.ncbi.nlm.nih.gov/nuccore/CP151802.1/), CP151803 (*Pedobacter sp.* FW305-3-2-15-E-R2A2—https://www.ncbi.nlm.nih.gov/nuccore/CP151803), CP152408 (*Xylophilus sp.* GW821-FHT01B05—https://www.ncbi.nlm.nih.gov/nuccore/CP152408), CP152407 (*Acidovorax sp.* FHTAMBA—https://www.ncbi.nlm.nih.gov/nuccore/CP152407) and CP152376 (*Rhodoferax sp.* GW822-FHT02A01—https://www.ncbi.nlm.nih.gov/nuccore/CP152376). The two plasmids developed in this study have been deposited to Addgene—pBWB507 (Addgene #209325) and pBWB514 (Addgene #209326). Datasets have been provided for all relevant figures—Dataset EV1 for Fig. EV1, Dataset EV2 for Fig. EV2, Dataset EV3 for Fig. 3, Dataset EV4 for Fig. 4C,D, and Dataset EV5 for Fig. 5. Dataset EV6 has been provided to compare differences between the genome of the Δ*pheA* strain and wild-type *E. coli* BW25113. Dataset EV7 provides all primers for this study. Dataset EV8 provides GenBank files for the plasmids developed in this study.

The source data of this paper are collected in the following database record: biostudies:S-SCDT-10_1038-S44320-024-00068-z.

## Peer review information

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

## Acknowledgements

The authors acknowledge the support of the Dean A. Richard Newton Memorial Professor Chair Funds. This work was supported by ENIGMA—Ecosystems and Networks Integrated with Genes and Molecular Assemblies (http://enigma.lbl.gov), a Science Focus Area Program at Lawrence Berkeley National Laboratory, supported by the United States Department of Energy, Office of Science, Office of Biological & Environmental Research under contract number DE-AC02-05CH11231. We acknowledge QB3 Genomics, University of California-Berkeley (Berkeley, CA), for sequencing support through RRID:SCR_022170. This work was supported by NIH S10 OD018174 Instrumentation Grant.

## Author contributions

**Bradley W Biggs**: Conceptualization; Data curation; Formal analysis; Validation; Investigation; Visualization; Methodology; Writing—original draft; Writing—review and editing. **Morgan N Price**: Conceptualization; Data curation; Software; Formal analysis; Investigation; Visualization; Methodology; Writing—original draft; Writing—review and editing. **Dexter Lai**: Investigation; Methodology. **Jasmine Escobedo**: Investigation; Methodology. **Yuridia Fortanel**: Investigation; Methodology. **Yolanda Y Huang**: Software; Formal analysis; Visualization. **Kyoungmin Kim**: Investigation; Methodology. **Valentine V Trotter**: Validation; Investigation. **Jennifer V Kuehl**: Data curation; Investigation. **Lauren M Lui**: Data curation; Software; Formal analysis; Validation. **Romy Chakraborty**: Validation; Investigation. **Adam M Deutschbauer**: Conceptualization; Formal analysis; Supervision; Investigation; Methodology; Writing—original draft; Project administration; Writing—review and editing. **Adam P Arkin**: Conceptualization; Supervision; Funding acquisition; Investigation; Methodology; Writing—original draft; Project administration; Writing—review and editing.

Source data underlying figure panels in this paper may have individual authorship assigned. Where available, figure panel/source data authorship is listed in the following database record: biostudies:S-SCDT-10_1038-S44320-024-00068-z.

## Disclosure and competing interests statement

The authors declare no competing interests.

# Expanded View Figures

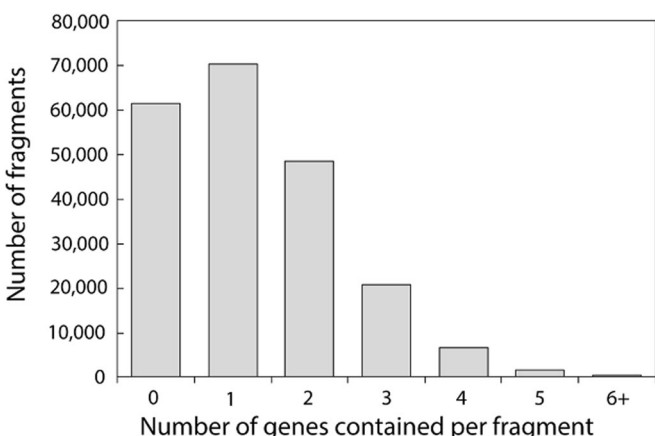

**Figure EV1.  Number of genes contained per fragment across all 11 libraries.**

The *x* axis shows the number of genes per fragment. The *y* axis shows the number of fragments per category.

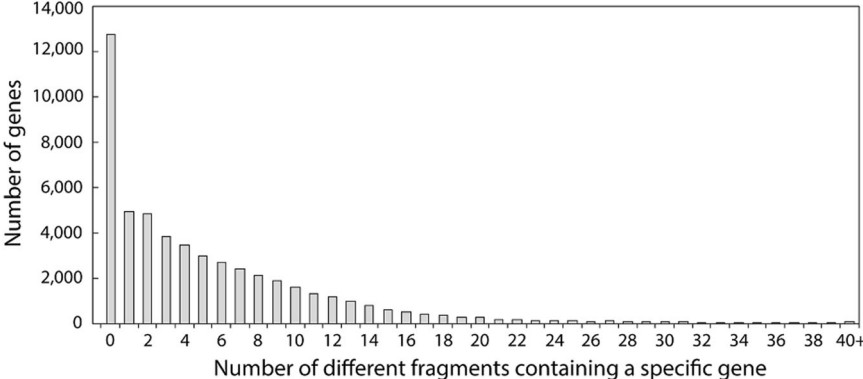

**Figure EV2.   Number of fragments covering each gene across all 11 libraries.**

The *x* axis shows the number of different unique fragments that cover a given gene. The *y* axis shows the number of genes covered per category (i.e., covered by 1 fragment, by 2 fragments, etc.).

