## [Peer Review File · Molecular Systems Biology]

High-throughput protein characterization by complementation using DNA barcoded fragment libraries

Bradley Biggs, Morgan Price, Dexter Lai, Jasmine Escobedo, Yuridia Fortanel, Yolanda Huang, Kyoungmin Kim, Valentine Trotter, Jennifer Kuehl, Lauren Lui, Romy Chakraborty, Adam Deutschbauer, and Adam Arkin

Corresponding author(s): Adam Arkin (aparkin@lbl.gov)

Review Timeline:

Submission Date:	8th May 24
Editorial Decision:	10th Jun 24
Revision Received:	1st Aug 24
Editorial Decision:	26th Aug 24
Revision Received:	5th Sep 24
Accepted:	9th Sep 24

Editor: Jingyi Hou

Transaction Report:

10th Jun 2024

Manuscript Number: MSB-2024-12412

Title: High-throughput protein characterization by complementation using DNA barcoded fragment libraries

Author: Bradley Biggs

Morgan Price

Dexter Lai

Jasmine Escobedo

Luis Fortanel

Yolanda Huang

Kyoungmin Kim

Valentine Trotter

Jennifer Kuehl

Lauren Lui

Romy Chakraborty

Adam Deutschbauer

Adam Arkin

Dear Prof Arkin,

Thank you for submitting your work to Molecular Systems Biology. We have now heard back from the three reviewers who agreed to evaluate your manuscript. As you will see from the reports below, the reviewers acknowledge the interest of the study. They raise, however, a series of concerns, which we would ask you to address in a major revision.

The reviewers' recommendations are relatively clear, so there is no need to reiterate the points listed below. In particular, after sending us their reviewer report, Reviewer #3 reached out to the editorial office and raised a concern regarding a preprint manuscript from your group (Y. Y. Huang, M. N. Price, A. Hung, O. Gal-Oz, D. Ho, H. Carion, A. M. Deutschbauer, A. P. Arkin, Functional screens of barcoded expression libraries uncover new gene functions in carbon utilization among gut Bacteroidales, *bioRxiv*, 2022.2010.2010.511384 (2022)). Reviewer #3 indicated that given the similarities between the current study and Huang et al 2022, it is crucial to compare and discuss both methods, particularly their differences and the respective pros and cons. Along those lines, Reviewer #2 also mentioned the current method needs to be better compared with other related methods.

All other issues raised by the reviewers need to be satisfactorily addressed as well. As you may already know, our editorial policy allows in principle a single round of major revision, and it is therefore essential to provide responses to the reviewers' comments that are as complete as possible.

On a more editorial level, we would ask you to address the following issues:

- Please provide a .docx formatted version of the manuscript text (including legends for main figures, EV figures and tables). Please make sure that the changes are highlighted to be clearly visible.
- Please provide individual production quality figure files as .eps, .tif, .jpg (one file per figure).
- Please provide a .docx formatted letter INCLUDING the reviewers' reports and your detailed point-by-point responses to their comments. As part of the EMBO Press transparent editorial process, the point-by-point response is part of the Review Process File (RPF), which will be published alongside your paper.
- Please note that all corresponding authors are required to supply an ORCID ID for their name upon submission of a revised manuscript.
- We replaced Supplementary Information with Expanded View (EV) Figures and Tables that are collapsible/expandable online (see examples in <http://msb.embopress.org/content/11/6/812>). A maximum of 5 EV Figures can be typeset. EV Figures should be cited as 'Figure EV1, Figure EV2' etc... in the text and their respective legends should be included in the main text after the legends of regular figures.

Additional Tables/Datasets should be labeled and referred to as Table EV1, Dataset EV1, etc. Legends have to be provided in a separate tab in case of .xls files. Alternatively, the legend can be supplied as a separate text file (README) and zipped together with the Table/Dataset file.

For the figures and tables that you do NOT wish to display as Expanded View figures, they should be bundled together with their legends in a single PDF file called *Appendix*, which should start with a short Table of Content. Each legend should be below

the corresponding Figure/Table in the Appendix. Appendix figures and tables should be referred to in the main text as: "Appendix Figure S1, Appendix Figure S2, Appendix Table S1" etc. See detailed instructions regarding expanded view here: <https://www.embopress.org/page/journal/17444292/authorguide#expandedview>.

-Before submitting your revision, primary datasets (and computer code, where appropriate) produced in this study need to be deposited in an appropriate public database (see [http://msb.embopress.org/authorguide - dataavailability](http://msb.embopress.org/authorguide-dataavailability) <https://www.embopress.org/page/journal/17444292/authorguide#dataavailability>).

The accession numbers and database should be listed in a formal "Data Availability" section (placed after Materials & Method) that follows the model below (see also <https://www.embopress.org/page/journal/17444292/authorguide#dataavailability>). Please note that the Data Availability Section is restricted to new primary data that are part of this study.

Data availability

-At EMBO Press we ask authors to provide source data for the main figures. Our source data coordinator will contact you to discuss which figure panels we would need source data for and will also provide you with helpful tips on how to upload and organize the files.

- Our journal encourages inclusion of *data citations in the reference list* to directly cite datasets that were re-used and obtained from public databases. Data citations in the article text are distinct from normal bibliographical citations and should directly link to the database records from which the data can be accessed. In the main text, data citations are formatted as follows: "Data ref: Smith et al, 2001". In the Reference list, data citations must be labeled with "[DATASET]". A data reference must provide the database name, accession number/identifiers and a resolvable link to the landing page from which the data can be accessed at the end of the reference. Further instructions are available at .

- We updated our journal's competing interests policy in January 2022 and request authors to consider both actual and perceived competing interests. Please review the policy <https://www.embopress.org/competing-interests> and update your competing interests if necessary.

Please use the heading "Disclosure statement and competing interests".

- All Materials and Methods need to be described in the main text. We would ask you to use 'Structured Methods', our new Materials and Methods format, which is mandatory for Methods and Articles with a strong methodological focus. According to this format, the Material and Methods section should include a Reagents and Tools Table (listing key reagents, experimental models, software and relevant equipment and including their sources and relevant identifiers) followed by a Methods and Protocols section in which we encourage the authors to describe their methods using a step-by-step protocol format with bullet points, to facilitate the adoption of the methodologies across labs. More information on how to adhere to this format as well as downloadable templates (.doc or .xls) for the Reagents and Tools Table can be found in our author guidelines: . An example of a Method paper with Structured Methods can be found here: .

-Regarding data quantification:

Please ensure to specify the name of the statistical test used to generate error bars and P values, the number (n) of independent experiments (please specify technical or biological replicates) underlying each data point and the test used to calculate p-values in each figure legend. Discussion of statistical methodology can be reported in the materials and methods section, but figure legends should contain a basic description of n, P and the test applied.

Graphs must include a description of the bars and the error bars (s.d., s.e.m.).

- Please provide a "standfirst text" summarizing the study in one or two sentences (approximately 250 characters, including space), three to four "bullet points" highlighting the main findings and a "synopsis image" (550px width and 400-600 px height, PNG format) to highlight the paper on our homepage.

Here are a couple of examples:

<https://www.embopress.org/doi/10.15252/msb.20199356>

<https://www.embopress.org/doi/10.15252/msb.20209475>

<https://www.embopress.org/doi/10.15252/msb.209495>

When you resubmit your manuscript, please download our CHECKLIST (<https://www.embopress.org/pb-assets/embo-site/EMBO%20Press%20Author%20Checklist-1642513524327.xlsx>) and include the completed form in your submission.

Please note that the Author Checklist will be published alongside the paper as part of the transparent process (<https://www.embopress.org/page/journal/17444292/authorguide#transparentprocess>).

If you feel you can satisfactorily deal with these points and those listed by the referees, you may wish to submit a revised version of your manuscript. Please attach a covering letter giving details of the way in which you have handled each of the points raised by the referees. A revised manuscript will be once again subject to review and you probably understand that we can give you no guarantee at this stage that the eventual outcome will be favorable.

I look forward to receiving the revised manuscript soon.

Kind regards,
Jingyi

Jingyi Hou, PhD
Scientific Editor
Molecular Systems Biology

We realize that it is difficult to revise to a specific deadline. In the interest of protecting the conceptual advance provided by the work, we recommend a revision within 3 months (8th Sep 2024). Please discuss the revision progress ahead of this time with the editor if you require more time to complete the revisions. Use the link below to submit your revision:

IMPORTANT: When you send your revision, we will require the following items:

1. the manuscript text in LaTeX, RTF or MS Word format
2. a letter with a detailed description of the changes made in response to the referees. Please specify clearly the exact places in the text (pages and paragraphs) where each change has been made in response to each specific comment given
3. three to four 'bullet points' highlighting the main findings of your study
4. a short 'blurb' text summarizing in two sentences the study (max. 250 characters)
5. a 'thumbnail image' (550px width and max 400px height, Illustrator, PowerPoint or jpeg format), which can be used as 'visual title' for the synopsis section of your paper.
6. Please include an author contributions statement after the Acknowledgements section (see <https://www.embopress.org/page/journal/17444292/authorguide>)
7. Please complete the CHECKLIST available at (<https://bit.ly/EMBOPressAuthorChecklist>). Please note that the Author Checklist will be published alongside the paper as part of the transparent process (<https://www.embopress.org/page/journal/17444292/authorguide#transparentprocess>).
8. When assembling figures, please refer to our figure preparation guideline in order to ensure proper formatting and readability in print as well as on screen:

See also figure legend guidelines: <https://www.embopress.org/page/journal/17444292/authorguide#figureformat>

9. Please note that corresponding authors are required to supply an ORCID ID for their name upon submission of a revised manuscript (EMBO Press signed a joint statement to encourage ORCID adoption). (<https://www.embopress.org/page/journal/17444292/authorguide#editorialprocess>)

Currently, our records indicate that the ORCID for your account is 0000-0002-4999-2931.

Link Not Available

*** PLEASE NOTE *** As part of the EMBO Press transparent editorial process initiative (see our Editorial at

<https://dx.doi.org/10.1038/msb.2010.72>), Molecular Systems Biology publishes online a Review Process File with each accepted manuscripts. This file will be published in conjunction with your paper and will include the anonymous referee reports, your point-by-point response and all pertinent correspondence relating to the manuscript. If you do NOT want this File to be published, please inform the editorial office at msb@embo.org within 14 days upon receipt of the present letter.

Reviewer #1:

The authors describe a method (Coaux-Seq) to advance experimental annotation of gene functions from diverse bacteria, by genetic complementation of *E. coli* auxotrophic hosts. This is an important topic as accurate gene annotations are key to understanding the metabolic capacities of organisms. Yet, annotating genes experimentally is a labour-intensive process, which the developed method may simplify. Biggs et al brings complementation with genomic libraries to a higher scale by making it compatible with barcode sequencing, enabling them to screen genomic libraries of 11 diverse species in 20 auxotrophic *E. coli* hosts (mainly aminoacid auxotrophs).

The method is very relevant as it has potential to be expanded to other genes and functions, of which they explore key experimental settings. However, this manuscript could be of more use to the community by presenting the validations and limitations of the technique more clearly and systematically (see comments below which would help improve readability). Additionally, several key takeaways could be best reflected in figures and tables and it would benefit by being more concise on writing (see comments).

Major comments

1. Considering the system has expected "true positive" proof-of-concept complementation hits (at least for the *E. coli* complementing library there should be a ground truth hit), the hit calling criteria using fitness could be benchmarked against true positive and true negatives already in the section Total Library Hits. E.g. Figure 2A just for the subset of *E. coli* genomic library as proof-of-concept, how does the fitness score plot when the library is transformed alone or as a mix of gDNA libraries? This could help contextualise and validate the cut-off fitness scores used.
2. As the method is based on sheared DNA, it would be great if the authors could show how many of the cloned 3 kb DNA pieces contain complete genes. It could include a histogram of genes encoded per fragment (this could be added to Figures S1-10 or shown generally in Figure 2 would be a good complement), as well as the distribution of coverage per gene.
3. Figure 3 could be more effective reflecting the overall efficiency of the pipeline, as this is the main point of the story (incorporating S15 information). E.g. a) could have a breakdown of how many expected hits per species to how were validated by Coaux-Seq. Note that Figure S15 does not indicate what replicates mean in the figure and also could show datapoints (and do please recheck in the rest of the figure caption).
4. Liquid selection was discarded from the screen as the output seemed biased towards a few barcodes. However, the nature of the experiments in S5-S7 is selecting for likely 1 or few complementing genes, so this could be expected. In S5-S7 it would be meaningful to have an additional column with the genes encoded on the plasmids as currently only the barcode is noted. E.g. are the most abundant barcodes from liquid selection plasmids containing the predicted complementing ortholog of Δ thrB (this should be clear at least in *E. coli* BW25113 into Δ thrB?) Are the other lowly seen plasmid expected to contain meaningful genes? This would help justifying that liquid strategy is not viable, if showing that are "false negatives" (and also after how many generations in liquid culture this bias is seen?). If the limitation on liquid arises when using multiple gDNA libraries together of expected uneven performance, this could be discussed. Considering that liquid vs solid selection could have an impact on scalability, addressing this rationally is important.
5. There are many sections with numbers commented on the text where it could be useful to visualise the data (e.g. codon adaptation index and RBS comparisons). As this data is also not in supporting at least could be present as figures (especially for section Expected Hit Recovery).

Minor:

1. Figure 1: Suggest changing Barseq image as the sanger chromatogram could be confusing as it is NGS (considering sanger is used on some cases or could be an alternative, to make the distinction clear).
2. For the cases where mutants "were auxotrophs by RB-TnSeq". How was this determined? Is fitness in defined media in pooled tests? This could be clearer.
3. Please clarify the strains in the captions of Figure S14. In general figure captions could be more descriptive (e.g. which type of replicates, strains genotypes, etc).
4. Figure 2a could have in parenthesis the amount of inducer (even if in x).
5. "These small faint colonies were observed even in control experiments run with red fluorescent protein (mRFP) in place of a genome fragment." Please explain why satellite colonies would be seen in the mRFP context? Is this a case of a positive selection example coencoded with mRFP?
6. Figure 3b the axis should have a legend and has a typo in "*E. coli*".
7. Criteria of expected unexpected is at points unclear e.g. Table S8 hisC complementing entries comments are very similar while one is considered expected and the other one is not (31% and 32% identity to HisC both operons however one expected

one unexpected)?

8. Figure S31 is cut in a way that second plate cannot be fully read.

9. The abstract could be more informative. The authors mention fragment libraries from 11 different bacteria. I assume the 41% recovery refers to the most successful one. But how successful was the least successful one? Are there huge differences between the libraries created? If so, why? I think this information is important for the reader to have in first place.

10. Also there may be the possibility that operons, when expressed partially may lead to functions not present in the original host bacterium, leading to artefacts. How do the authors deal with this? Did they come across such artefacts? Would they be able to recognise them? If so it would be great to provide a list, as they may be useful e.g. for creating enzymes with novel/improved functions.

Reviewer #2:

The Authors present a method for experimentally annotating unknown bacterial proteins with functions. The method takes an advantage of auxotrophic *Escherichia coli* strains in which foreign DNA fragments are screened for activities complementing the auxotrophies. The method addresses a major limitation. Incompleteness and inaccuracies in protein functional annotation indeed cause severe challenges in system wide methods like genome-scale metabolic modelling. Since computational protein functional annotation methods are yet inherently limited to available data on already annotated proteins, improving the data availability calls for new experimental high-throughput functional annotation methods. Thus, I find the method relevant and interesting for the audience of Molecular Systems Biology. However, I have some concerns on the method and how it is presented in the manuscript. Please, find my detailed concerns below.

The method is insufficiently compared to other means for experimentally annotating protein functions in single species or across species. State of the art is not presented either in Introduction or in Discussion.

It remains also questionable whether this method can really be considered a high-throughput method as the clones were manually picked from plates and the number of available auxotrophies limits the protein functions that can be identified.

Furthermore, expressing and screening many overlapping fragments appeared important for reliable function identifications.

Is it beneficial to use this global method also when the interest is on a single species instead of first predicting genes and focusing the cloning and screening to those like the 14-22 candidate genes identified here for complementing the 19 auxotrophies? Above 40% of sequence similarity to annotated proteins the annotation could be doable using computational methods. Here only 11 proteins with lower similarity were annotated out of 11 genomes.

Insert fragments were large. It remains unclear how often they contained a couple of genes instead of a single (based on gene predictions). Could the effect of an additional gene mask the completion of the auxotrophy? How many insert fragments actually need to be screened for an average genome size? Is the genomic DNA shearing optimal?

Bias to top candidates in liquid cultures depends on the growth rate difference between the clones. Perhaps the liquid growth assay could be performed if the maximum growth rate was reduced e.g., by the choice of the carbon source or other modification to the chemical environment.

I would recommend removing the summary section from the results. It did not improve clarity and I found it repetitive.

I have the following concerns on the formatting:

Past tense should be consistently used when reporting the work done.

Numbers up to twelve should be written out

Figure referencing calls for a revision like on page 6 the main text contains figure legend text.

Units for fitness are missing

"complimenting" in the abstract should be complementing

Clarification for when statistical testing was used for confirming significance is occasionally missing.

Figure 3b *E. coli* should be *E. coli*

In addition I have a couple of minor concerns:

By definition a minimal medium contains only the necessary nutrients for growth, is this the case here (page 4)?

I would expect not knockouts but complementing genes on page 7.

Reviewer #3:

Manuscript ID: MSB-2024-12412

Title: High-throughput protein characterization by complementation using DNA barcoded fragment libraries

Authors: Bradley W. Biggs et al.

This manuscript describes the development of a new high-throughput barcoded plasmid system named "Coaux-Seq."

The authors first developed the expression vector with some modifications, including the barcode system to the original vector, pBbA2c-RFP(Addgene #35326), to create high-throughput shotgun expression libraries.

The authors conducted functional screening on 11 diverse bacterial genomic fragment libraries. They identified and functionally screened plasmids carrying functionally complementing fragments from 20 different auxotrophs of *E. coli* on the M9 minimal medium. The results of this screening provide a comprehensive understanding of the 'Coaux-Seq' system's capabilities.

This manuscript has valuable information for the readers and is acceptable for publication after carefully considering the comments below.

Major points;

1) The numerical information on the breakdown of candidate gene groups in the case studies "Expected recovery," "Diverged hits," and "Identifying novel enzymes and transporters" is very confusing. It would be easier to understand if it were shown in a flowchart or summarized in a simple table.

2) Is it unnecessary to examine the stability of the individual fragments in the established library? I assume that the individual plasmids are not separated and purified but stored in a pool of plasmids, but if this is the case, it is also necessary to discuss how much amplification the cloned plasmids of individual fragments can withstand during the repeated use stage. Moreover, in the current analysis, the authors primarily employed functional complementation. This approach, which relies on the similarity of the genes encoded in the complementary fragments for analysis and interpretation, can be viewed as a validation of the results of homology analysis. It is particularly intriguing as it has the potential to identify fragments encoding genes with very low homology or no homology, such as in convergent evolution. Could the authors delve deeper into such cases?

Minor comments;

1) Page 4, line 18, "oriT" is mentioned in the text. p15A replication origin information is important information about the copy number of this vector, but what is the advantage of the oriT information for this purpose? If the authors have a clear, advantageous way of using oriT for the same purpose, I think it is generous to mention the benefit of oriT.

2) Page 8, "Expected Hit Recovery" section, line 12, the definition of "t0" is missing. "t0" means the reads on the LB rich medium recovery. I think it should be clear in Figure 1 or the materials and methods part.

3) Page 5, Fig 2 a), I think there is no reason to show "not significance" by grey box and grey spots. Just by grey spots, it might be simple. Is there any other reason to distinguish two ways? Or the authors need to add the statement (page 6, line 9, (fitness > 5 and z-like test statistic > 4)) to the legend.

4) Page 8, "Expected Hit Recovery," 13, "remaining 152 genes covered by a fragment, only 132 of them were seen at t0." why "remaining"? Out of 203 expected complementing genes, 75% means 152 genes, I read. Is this correct? If so, I think "remaining" is unreasonable. Out of 152, 132 were seen by sequence reads at t0. So, 20 fragments were not observed at t0. This depends on the number of long-read identifications of the correspondence between the fragments and barcodes, the size of the initial library, and the total genome size of 11 strains. What is the estimated coverage of the library of the genomes analyzed? This situation might be reasonable if functional screening were used. "2% vs. 50%" is also confusing for me. These numbers came from where?

5) Page 9, line 8, and supplementary information Fig. S15, in the main text, shows 72% but in the Sup. Fig. S15 shows the ratio. Unifying these units is clearer.

6) Page 9, line 11, and supplementary information Fig. S15, species names in the Fig. S15 X-axis and the main text are different. Unifying those names is easier.

7) Page 9, second paragraph line 26, "poorly (35% recovery)", it is not easy for readers to identify :S, koreensis" on Fig. S15.

8) Page 9, second paragraph line 32, "Overall, the RBS strength prediction did not correlate with individual protein complementation success.", I think the plasmid is recoverable and is it possible to check the expression individually?

9) Page 10, "Diverged hits" section, "Additionally, we categorized a set of "diverged" hits (10 in total) that could have been expected to complete the missing function, but with caveats to high-confidence assignment. Hits were considered "diverged" if they were similar to experimentally-characterized proteins that have the missing activity (and not more similar to proteins known to have other functions instead), but are <40% identical on an amino acid basis to any characterized protein with that activity in curated databases". For me, it is not easy to understand. "similar" means structural similarity by sequence similarity analysis? If so, "<40%" means not a similar sequence, probably in the amino acids sequence. Is this correct? This situation seems interesting, and it might be possible to consider this as the case of convergent evolution. Why didn't you try to compare the estimated 3D structure by AlphaFold? If these connect to Figs. 4 and 5, re-arrange the "Diverged hits" and "Identifying novel enzymes and transporters" sections.

10) Page 10, "Identifying novel enzymes and transporters" section, line 10, "cross feeding" and Sup. Figs 31-33, can the secreted metabolites reach the neighboring spots even at such a distance? This is very interesting, but can we call it cross-feeding? It is possible to mix isolated strains and see the growth of each cell more quantitatively, but is this not considered necessary?

11) Page 13 last line, "Moreover, RB-TnSeq data for LRK54_RS05660 does not show auxotrophic phenotypes" is not surprising because of the redundancy of genes. Did you check the genes in the same strain that are similar?

12) Page 14, line 3 and 5, LRK54_RS05660 is the target to be analyzed for the complementation to DmetC but LRK54_RS17455 is mentioned in line 5. Is this just a mistake or my misunderstanding?

13) On page 14, line 19, I think this point is important. Is there any structure-based discussion about this point? Is there a difference at the catalytic active center or other structural domains for other functional possibilities?

14) Page 15, line 7, "Coaux-Seq successfully uncovered novel biochemical function." I think
It is generally understood that functional complementation fragment analysis is an essential element and that barcode technology enables quantitative analysis of this using a pooled library. However, I feel a little uncomfortable when it is vigorously explained that Coaux-Seq can do it.

**Lawrence Berkeley
National Laboratory**

Environmental Genomics & Systems Biology Division

The reviewers' recommendations are relatively clear, so there is no need to reiterate the points listed below. In particular, after sending us their reviewer report, Reviewer #3 reached out to the editorial office and raised a concern regarding a preprint manuscript from your group (Y. Y. Huang, M. N. Price, A. Hung, O. Gal-Oz, D. Ho, H. Carion, A. M. Deutschbauer, A. P. Arkin, Functional screens of barcoded expression libraries uncover new gene functions in carbon utilization among gut Bacteroidales, *bioRxiv*, 2022.2010.2010.511384 (2022)). Reviewer #3 indicated that given the similarities between the current study and Huang et al 2022, it is crucial to compare and discuss both methods, particularly their differences and the respective pros and cons. Along those lines, Reviewer #2 also mentioned the current method needs to be better compared with other related methods.

We would be happy to clarify the differences between Coaux-Seq and other related methods, including our own like Boba-seq (Huang et al., *bioRxiv*, 2022). With respect to Boba-seq, which we cite upfront in our manuscript, while both techniques utilize DNA barcoding and synthetic vector driven overexpression of genetic fragments, the genetic content and selection context differs significantly (not to mention the entirety of the results). Huang et al. focuses on the anaerobic gut microbiome, specifically carbohydrate metabolism and environmental stressors (antibiotics), generating an anaerobic relevant expression vector and screening in the context of the model gut microbiome host *Bacteroides thetaiotaomicron*. Huang et al. builds on Dub-Seq (Mutalik et al., *Nature Communications*, 2019), in which *E. coli* genomic fragments were overexpressed in *E. coli* in the context of different medium conditions such as metal or antibiotic stress. Both previous works (Dub-seq, Boba-seq) utilize gain-of-function screening, in contrast to loss-of-function screens like those based on transposon mutagenesis (e.g., RB-TnSeq). In each, NGS ascertained barcode frequency indicates if contained material positively or negatively correlated with cellular fitness in a given context.

Coaux-Seq presents a new expression vector, suitable for use with *E. coli* Keio knockout strains based on antibiotic resistance compatibility. This new vector also contains an oriT to enable conjugation and has greater library portability because of the type IIS restriction sites flanking the fragment + barcode, allowing for long sequencing mapped libraries to be re-cloned into new expression vectors without PCR. The screening context of this work differs from previous works (introduced stresses and varied medium conditions), assaying instead in the context of auxotrophic genetic knocks outs. Thus, the present work more directly connects protein function to a missing biochemical activity and builds on the historical functional complementation assays. Further distinguishing the work, and representing a non-trivial progression, the genetic material tested here is predominantly from recently isolated soil bacteria, contrasting the *E. coli* only and gut bacteria focus of prior works. Coaux-Seq assays if DNA from diverse hosts, including different orders, classes (Sphingomonas), and phyla (Bacillus, Bacteroides, Pedobacter) can function in *E. coli*. Our present work also demonstrates that the liquid assays utilized in the Boba-seq work are likely not suitable when testing genetic material from diverse species. Finally, the results themselves are entirely different (e.g., the 53 proteins with first experimental evidence of function).

To clarify these distinctions in the manuscript, we modified the first paragraph of the discussion to the following,

“Coaux-Seq provides another important tool for functional genomics and advancing high-throughput annotation of genetic function. Coaux-Seq builds on existing gain-of-function tools like Dub-seq (Mutalik et al, 2019) and Boba-seq (Huang et al, 2022), providing a new expression vector and introducing diverse bacteria and auxotrophy as a testing context. This method compliments loss-of-function tools like RB-TnSeq (Price et al, 2018; Wetmore et al, 2015) and various CRISPRi strategies (Enright et al, 2024). Similar to how RB-TnSeq dramatically improved the throughput of transposon-based loss-of-function experiments that have been employed for many years (Hensel et al, 1995; Cain et al, 2020), Coaux-Seq significantly improves the throughput of gene complementation studies, which have been utilized for even longer (Fincham, 1968). These modern techniques owe their improved throughput over historical characterization approaches to advances in multiplexing and next generation sequencing (NGS), as covered thoroughly by Gray and coworkers (Gray et al, 2015). Coaux-Seq accesses this benefit by creating DNA-barcoded overexpression libraries from randomly sheared genomic fragments and then mapping with long-read sequencing technology (PacBio) to enable repeated assaying of contained genes to complete missing biochemical function. By utilizing BarSeq, which leverages primers containing Illumina adapters and only requires short sequencing read length, it also is cost-effective. Ultimately, this cost-effective method not only provided the first experimental validation for the activity of 53 proteins, but also uncovered novel function. Figure 9 provides a flow-chart style summary of the findings of this study including hit identification, hit categorical breakdown, expected hit recovery, and novel findings.”

--

Reviewer #1:

The authors describe a method (Coaux-Seq) to advance experimental annotation of gene functions from diverse bacteria, by genetic complementation of *E. coli* auxotrophic hosts. This is an important topic as accurate gene annotations are key to understanding the metabolic capacities of organisms. Yet, annotating genes experimentally is a labour-intensive process, which the developed method may simplify. Biggs et al brings complementation with genomic libraries to a higher scale by making it compatible with barcode sequencing, enabling them to screen genomic libraries of 11 diverse species in 20 auxotrophic *E. coli* hosts (mainly amino acid auxotrophs).

The method is very relevant as it has potential to be expanded to other genes and functions, of which they explore key experimental settings. However, this manuscript could be of more use to the community by presenting the validations and limitations of the technique more clearly and systematically (see comments below which would help improve readability). Additionally, several key takeaways could be best reflected in figures and tables and it would benefit by being more concise on writing (see comments).

Major comments

1. Considering the system has expected "true positive" proof-of-concept complementation hits (at least for the *E. coli* complementing library there should be a ground truth hit), the hit calling criteria using fitness could be benchmarked against true positive and true negatives already in the section Total Library Hits. E.g. Figure 2A just for the subset of *E. coli* genomic library as proof-of-concept, how does the fitness score plot when the library is transformed alone or as a mix of gDNA libraries? This could help contextualise and validate the cut-off fitness scores used.

This is an excellent question. With respect to the use of the *E. coli* genomic fragment library as a true positive, we have added the following discussion to the start of the “Expected Hits Recovery” section. Included in it is the number of *E. coli* genes that returned a hit (19/20). The only instance that did not

provide a hit was for $\Delta cysH$. Interestingly, 22 different mapped inserts matched *E. coli* CysH, meaning there were numerous chances to find a successful fragment. Yet, this transformation showed very few transformed *cysH* containing barcodes from any source genome (33) and very few of the *E. coli* inserts were successfully transformed (14% observed at t_0). Examining this further, it has been observed that $\Delta cysH$ strains can accumulate mutations in upstream pathway enzymes (*cysA*, *cysD*, *cysN*) to avoid futile ATP utilization (Gillespie et al., 1968). Below is the text that has been added,

“We initially assessed if each *E. coli* gene could successfully complement its own loss and be identified in our assay. We observed successful complementation for 19/20 (95%) auxotrophs, including *CysA*. The only absent complementation was *CysH*, which both had a low transformation efficiency (few barcodes observed at t_0) and a potential complicating factor of futile ATP usage stress (Gillespie *et al.*, 1968). More broadly, ...”

Addressing the consideration of using *E. coli* as a ground truth or benchmark, for *E. coli* complementing fragments in the individual library experiments the median fitness value was 10.3 and in the mixed 11 library experiments the value was 12. However, our expectation would be that these *E. coli* fragments provide much higher fitness values than those from other organisms because of superior expression (native promoters and RBSs) and evolutionary optimization for function in *E. coli*. Meaning, using these values as cutoffs would likely be too stringent and cause us to increase our false negative rate.

With respect to the concern that the fitness value may not work equally in the single genome and mixed 11 genome experiments, using our fitness cutoff value of 5, if an expected hit barcode was observed at t_0 the success rate was very similar between single genome and mixed genome experiments (88% vs. 84% recovery). If we increased our fitness cutoff from 5 to 6, our number of observed high-confidence hits for expected hits reduces from 83 to 80 and for all others reduces from 77 to 42. This equates to a likely reduction in false positives but also of true positives. To provide context for the reader, we have adapted Figure S30 to make new Figure 5 (shown below). This figure shows the distribution of fitness scores and fraction supported by overlapping fragments per category, and now includes the other untested expected hits. For ease of reading, we have included the average score of each category in bold to the left of the category title, along with the percentage of hits that have an overlapping fragment. This updated figure is shown below.

Categories of Hits

Lastly, to add this context for the reader, we have included the following at the end of the discussion section,

“On this note, future studies may consider adjusting their fitness value cutoffs. If we had increased our fitness cutoff from 5 to 6, we would have observed a decrease in unexpected hits, which likely contain false positives (from 75 to 40), but also a slight decrease in recovered expected hits (from 83 to 80). The optimum cutoff may be context dependent.”

2. As the method is based on sheared DNA, it would be great if the authors could show how many of the cloned 3 kb DNA pieces contain complete genes. It could include a histogram of genes encoded per fragment (this could be added to Figures S1-10 or shown generally in Figure 2 would be a good complement), as well as the distribution of coverage per gene.

We have created two additional Figures (now Figures EV1 and EV2, shown below) that show the number of genes contained per fragment across all 11 libraries (Figure EV1) and the distribution of coverage per gene (Figure EV2). In addition, all of the figures for the 11 individual libraries (Appendix Figures S1-11) have now been updated to contain these same kinds of histograms. Lastly a supplemental table has been added with this information in case that is advantageous for the reader (Table EV2). One minor note, some genes are not covered for their entire reading frame, yet are still able to function.

3. Figure 3 could be more effective reflecting the overall efficiency of the pipeline, as this is the main point of the story (incorporating S15 information). E.g. a) could have a breakdown of how many expected hits per species to how were validated by Coaux-Seq. Note that Figure S15 does not indicate what replicates mean in the figure and also could show datapoints (and do please recheck in the rest of the figure caption).

We have updated Figure 3 (now Figure 4) to include S15 (as panel d) and to include a breakdown of how many expected hits per genome there were and how many of the expected hits were validated per genome (as panel b). In panel b) we have the fraction of successful expected hits per genome with the number above the bar. The legend in Figure 4a is carried throughout. In addition, we have updated the description of the error bars for Figure 4d (formerly Figure S15). They represent the 90% confidence intervals (binomial test). Below is the updated figure,

a) Expected hits per library

c) Promoter and gene orientation of hits

b) Fraction of success for expected hits

d) Fraction of success for inserts detected at t₀

Lastly, we have rechecked all of our figure captions. Thank you for this suggestion.

4. Liquid selection was discarded from the screen as the output seemed biased towards a few barcodes. However, the nature of the experiments in S5-S7 is selecting for likely 1 or few complementing genes, so this could be expected. In S5-S7 it would be meaningful to have an additional column with the genes encoded on the plasmids as currently only the barcode is noted. E.g. are the most abundant barcodes from liquid selection plasmids containing the predicted complementing ortholog of Δ thrB (this should be clear at least in *E. coli* BW25113 into Δ thrB?) Are the other lowly seen plasmid expected to contain meaningful genes? This would help justifying that liquid strategy is not viable, if showing that are "false negatives" (and also after how many generations in liquid culture this bias is seen?). If the limitation on liquid arises when using multiple gDNA libraries together of expected uneven performance, this could be discussed. Considering that liquid vs solid selection could have an impact on scalability, addressing this rationally is important.

First, we have updated Tables S5-7 (now Tables S6-8) to contain gene information in an additional column as suggested. We have bolded the expected to be complementing genes. As anticipated, the top read count barcodes/fragments contain expected complementing genes. Importantly, however, from our library mapping we know that for BW25113_RS00015 (*E. coli* thrB, Table S6) there are 15 different fragment that contain the gene (10 sense, 5 antisense), for BDW16_RS16080 (*S. koreensis* homoserine kinase, Table S7) there are 9 different fragments that contain the gene (4 sense, 5 antisense), and for TK06_RS20845 (*P. fluorescens* homoserine kinase, Table S8) there are 17 different fragment that contain the gene (8 sense, 9 antisense). Yet, we only observe one or two of these fragments at most in these liquid culture assays. In contrast, for the solid medium assays we observe 14 of the fragments for BW25113_RS00015, 4 fragments for BDW16_RS16080, and 10 different fragments TK06_RS20845, clearly indicating the reduction of fragments that are observed by liquid screening. Thus, the issue is not necessarily the occurrence of low lying "false negatives", but that several known true positives are not observed in liquid selection. Because these other known true positives aren't among even the low-lying reads, modest increases or decreases in the number of generations are unlikely to have a significant impact.

To clarify this point in the text we have added the following,

"When selection was conducted for initial experiments in liquid culture, the top two barcodes accounted for >99% of all reads in all three initial cases tested (**Tables S6-8**). While top barcodes corresponded to expected genes, these assays utilizing the individual libraries for *E. coli*, *S. koreensis*, and *P. fluorescens* FW300-N2E2 transformed into Δ thrB delivered 1, 1, and 2 of the possible 15, 9, and 17 expected gene fragments, respectively. In contrast, solid medium selection for the same libraries in Δ thrB delivered 14, 4, and 10 of the expected gene fragments, indicating a clear bias introduced by liquid selection."

5. There are many sections with numbers commented on the text where it could be useful to visualise the data (e.g. codon adaptation index and RBS comparisons). As this data is also not in supporting at least could be present as figures (especially for section Expected Hit Recovery).

We have now included a new supplemental figure (new Appendix Figure S15, shown below) that provides a plot of GC content, RBS strength, and codon adaptation index, comparing successful and unsuccessful expected hits.

We have also introduced new text in the “Expected Hit Recovery” section to summarize these findings,

“**Appendix Figure S15** provides plots comparing successful and unsuccessful cases with respect to their GC content, predicted RBS strength, and codon adaptation index. As can be seen from the plot, successful and unsuccessful cases have almost completely overlapped values, and thus no obvious trend can be ascertained.”

Minor:

1. Figure 1: Suggest changing Barseq image as the sanger chromatogram could be confusing as it is NGS (considering sanger is used on some cases or could be an alternative, to make the distinction clear).

We appreciate this suggestion, and have updated Figure 1 to distinguish both steps of sequencing that are primarily PacBio and then Illumina to look distinct from Sanger.

2. For the cases where mutants “were auxotrophs by RB-TnSeq”. How was this determined? Is fitness in defined media in pooled tests? This could be clearer.

We searched for the phrase “were auxotrophs by RB-TnSeq” and were unable to find exactly where in the manuscript the reviewer is referring to. However, we agree that clarifying the relevance of cross-referenced RB-TnSeq data is important for the reader. Accordingly, we have updated the wording of its first call out in the text to explain RB-TnSeq briefly and how the data was used (new text underlined).

“Close homologs of AAF35_21465 are essential proteins, and it is 45% identical to HMPREF1058_RS13970 from *Phocaeicola vulgatus* CL09T03C04, which is cofit with chorismate synthase (AroC) in RB-TnSeq fitness data (Surya Tripathi, personal communication). As RB-TnSeq utilizes transposon mutagenesis to disrupt genes across a host organism (loss-of-function), correlated fitness often indicates that the two genes have related function (Price *et al*, 2018).”

In another reference to RB-TnSeq, perhaps the one indicated by the reviewer, we have updated the wording to mention what is meant by an auxotrophic phenotype (provision of fitness > 0) and the context in which this was tested (minimal glucose medium). Genes with auxotrophic phenotypes will show a growth disadvantage in these conditions. The new text reads,

“Moreover, RB-TnSeq data for LRK54_RS05660 does not show auxotrophic phenotypes (Hira Lesea, personal communication) leaving unclear its function. Specifically, RB-TnSeq assays with a pool of mutants in minimal glucose medium found fitness values greater >0 for this gene, which indicates that mutants of this gene had no growth disadvantage (thus, could not be auxotrophic).”

We have updated an additional RB-TnSeq call out in the following manner to clarify meaning,

“Interestingly, neither FHTAMBA nor GW101-3H11 has a strong candidate for MetC, and the best candidate in the GW101-3H11 strain (Ac3H11_34) is more similar to methionine gamma-lyases and is not important for growth in minimal media (all RB-TnSeq fitness >0, indicating no disadvantage for mutants of this gene) (Price *et al*, 2018).”

3. Please clarify the strains in the captions of Figure S14. In general figure captions could be more descriptive (e.g. which type of replicates, strains genotypes, etc).

Thank you for raising this concern. We have added the follow to the caption,

“This test was specifically done with three different transformations of an individual genome fragment library into an auxotrophic strain. The lightest gray bar is for the transformation of the *B. thetaiotaomicron* library into Δ aroA. The medium shade gray bar is for the transformation of the *B. subtilis* library into Δ thrB. The black bar is for the transformation of the *B. subtilis* library into Δ aroA. Thus, we tested two different libraries and two different genetic knockout backgrounds to determine the optimum.”

In addition, we have looked through all the other captions to add detail throughout the manuscript and appendix.

4. Figure 2a could have in parenthesis the amount of inducer (even if in x).

We have updated the figure to include 1x aTc and 5x aTc where appropriate.

5. "These small faint colonies were observed even in control experiments run with red fluorescent protein (mRFP) in place of a genome fragment." Please explain why satellite colonies would be seen in the mRFP context? Is this a case of a positive selection example coencoded with mRFP?

These reasons for seeing small faint colonies for the mRFP condition are those described in the text. Namely, nutrient carry over from previous culturing or nutrient impurities contained in the agar. It is also possible, as in the case for pheA, that spontaneous non-enzymatic reactions can provide modest supplementary activity for the lost enzymatic activity. The mRFP cloning does not include the possibility of co-encoded fragments. Also, to clarify, these are not “satellite colonies” in the sense of forming around a larger healthy colony, but very faint and evenly distributed across the plate.

6. Figure 3b the axis should have a legend and has a typo in "E coil".

Thank you, we have corrected this typo (now Figure 4). We have added a label to the y-axis. The legend is the cartoon to the left, where the upper gray box matches antisense (upper portion of bar plot) and the lower green box matches sense. We have added a description to the caption to help clarify this.

7. Criteria of expected unexpected is at points unclear e.g. Table S8 hisC complementing entries comments are very similar while one is considered expected and the other one is not (31% and 32% identity to HisC both operons however one expected one unexpected)?

We apologize for any confusion. This labeling is because we applied a systematic criterion for defining “expected hit,” and one of these cases is categorized as expected hit by this criterion and the other is not. Yet, with manual inspection, we *could* have expected it. As defined in the text, if a gene was >40% homologous to an experimentally validated enzyme with the appropriate function and was the only option in the genome, then it was considered an expected hit.

For the examples the reviewer references, the Diverged Hits table (Appendix Table S7) does list two diverged HisC genes, both in histidine operons. One is expected and the other isn't. These two genes are both sufficiently diverged that they would not be considered as high-confidence candidates based on pairwise similarity. However, AAF19_05770 is considered expected because of a hit to a TIGRFams for HisC. The other, AAGF34_01100 from *Rhodospirillum rubrum* sp. GW822-FHT02A01, was also found by TIGRFam, and was considered a high-confidence candidate, but GapMind also found two other high-confidence candidates in this genome. One of these (AAGF34_11740) was selected as the expected hit because it was over 50% identical to previously-characterized hisC proteins (PP_0967 and Pf6N2E2_3251). Thus, because of higher amino acid identity AAGF34_11740 was labeled the expected hit and AAF19_05770 was not.

We have updated the text of the diverged hits section to include this information to help clarify this point. In addition, the overview figure (Figure 9) hopefully helps clarify the categories of the hits.

8. Figure S31 is cut in a way that second plate cannot be fully read.

The cutoff portion of the right plate reads M9 5x aTc (description of plate). Apologies, this was just how the photo was taken and not an intentional cropping. A second picture of the same plates is shown here, which shows more of the plate but has a significant glare, which is why it wasn't used.

9. The abstract could be more informative. The authors mention fragment libraries from 11 different bacteria. I assume the 41% recovery refers to the most successful one. But how successful was the least successful one? Are there huge differences between the libraries created? If so, why? I think this information is important for the reader to have in first place.

We apologize for the lack of clarity; we are working within the 175-word limit. The 41% recovery is across all genomes. In addition to updating and introducing new figures in the manuscript showing this data (namely what is now Figure 4), we have updated this sentence of the abstract to provide additional clarity.

“We recovered 41% of expected hits, with effectiveness ranging per source genome, and observed success even with distant *E. coli* relatives like *Bacillus subtilis* and *Bacteroides thetaiotaomicron*.”

If given an editorial exception to exceed the 175-word limit, we could include additional information such as “...10% recovery of expected genes for *Xylophilus sp.* to 95% recovery for *E. coli*, 41% averaging across all genomes.”

One of the primary factors for determining recovery efficiency was cloning efficiency. The two genomes with the lowest recovery % (*Pedobacter sp.* and *Xylophilus sp.*) correspond to those with the lowest cloning efficiency in the library creation step (fewest mapped fragments per genome size). The cloning of these libraries was attempted multiple times, and it is unknown as to why the efficiency was lower than for other genomes, as all were conducted in the same manner. Because of this, the efficiency numbers require some nuance in interpretation, which might be difficult to capture adequately in the abstract, regardless of an allowance for a slightly higher word limit.

10. Also there may be the possibility that operons, when expressed partially may lead to functions not present in the original host bacterium, leading to artefacts. How do the authors deal with this? Did they come across such artefacts? Would they be able to recognise them? If so it would be great to provide a list, as they may be useful e.g. for creating enzymes with novel/improved functions.

Partial operons could represent a source of variation in our results. Our primary means of addressing this variable was to generate libraries where each gene is covered by multiple different fragments and in slightly different contexts (i.e., different genes including alongside it, both orientations). Some genes have overlapping open reading frames with neighboring genes and cannot be fully decoupled.

With respect to greater-than-native expression, our use of a synthetic promoter could lead to the expression of genes that are not expressed in their native environment. Because of this synthetic expression, we may observe activity from genes that do not perform the same function in their host organism, or where low-level promiscuous function is observed in our assay because of overexpression. We note in what is now the supplementary note in the Appendix that the putative cystathionine gamma-lyase from *Rhodanobacter denitrificans* FW104-10B01 found in the context of $\Delta metB$ may be an example. This is the only case of which we are aware of among our findings.

Reviewer #2:

The Authors present a method for experimentally annotating unknown bacterial proteins with functions. The method takes an advantage of auxotrophic *Escherichia coli* strains in which foreign DNA fragments are screened for activities complementing the auxotrophies. The method addresses a major limitation. Incompleteness and inaccuracies in protein functional annotation indeed cause severe challenges in system wide methods like genome-scale metabolic modelling. Since computational protein functional annotation methods are yet inherently limited to available data on already annotated proteins, improving the data availability calls for new experimental high-throughput functional annotation methods. Thus, I find the method relevant and interesting for the audience of Molecular Systems Biology. However, I have some concerns on the method and how it is presented in the manuscript. Please, find my detailed concerns below.

The method is insufficiently compared to other means for experimentally annotating protein functions in single species or across species. State of the art is not presented either in Introduction or in Discussion.

As mentioned above in our response to the editor's note at the top, we have added information to the discussion to reference state of the art, including a reference to the review by Gray et al., which thoroughly covers competing technologies.

“Coaux-Seq provides another important tool for functional genomics and advancing high-throughput annotation of genetic function. Coaux-Seq builds on existing gain-of-function tools like Dub-seq (Mutalik et al, 2019) and Boba-seq (Huang et al, 2022), providing a new expression vector and introducing diverse bacteria and auxotrophy as a testing context. This method compliments loss-of-function tools like RB-TnSeq (Price et al, 2018; Wetmore et al, 2015) and various CRISPRi strategies (Enright et al, 2024). Similar to how RB-TnSeq dramatically improved the throughput of transposon-based loss-of-function experiments that have been employed for many years (Hensel et al, 1995; Cain et al, 2020), Coaux-Seq significantly improves the throughput of gene complementation studies, which have been utilized for even longer (Fincham, 1968). These modern techniques owe their improved throughput over historical characterization approaches to advances in multiplexing and next generation sequencing (NGS), as covered thoroughly by Gray and coworkers (Gray et al, 2015). Coaux-Seq accesses this benefit by creating DNA-barcoded overexpression libraries from randomly sheared genomic fragments and then mapping with long-read sequencing technology (PacBio) to enable repeated assaying of contained genes to complete missing biochemical function. By utilizing BarSeq, which leverages primers containing Illumina adapters and only requires short sequencing read length, it also is cost-effective. Ultimately, this cost-effective method not only provided the first experimental validation for the activity of 53 proteins, but also uncovered novel function. Figure

9 provides a flow-chart style summary of the findings of this study including hit identification, hit categorical breakdown, expected hit recovery, and novel findings.”

It remains also questionable whether this method can really be considered a high-throughput method as the clones were manually picked from plates and the number of available auxotrophies limits the protein functions that can be identified. Furthermore, expressing and screening many overlapping fragments appeared important for reliable function identifications.

We would like to clarify that clones were not manually picked from plates. Instead, plates were scraped together and pooled, allowing us to readily assay 11 different libraries at once. Moreover, automation tools (e.g., colony pickers) could rapidly accelerate related workflows if desired. Available auxotrophies do constrain what protein functions can be assayed, although multigene knock out strategies could extend the different biochemistries available as auxotrophies. Regardless, accurate annotation of core metabolisms (for which auxotrophies are generally available) is still an acute need. Lastly, it is true that improved cloning efficiency, and thus the presence of many overlapping fragments, benefits identification, which is a takeaway of this workflow establishing manuscript.

Is it beneficial to use this global method also when the interest is on a single species instead of first predicting genes and focusing the cloning and screening to those like the 14-22 candidate genes identified here for complementing the 19 auxotrophies? Above 40% of sequence similarity to annotated proteins the annotation could be doable using computational methods. Here only 11 proteins with lower similarity were annotated out of 11 genomes.

If one is interested in only single species and a single or few activities, then high-throughput methods will often not be the best choice. However, for newly isolated hosts and from a systems biology perspective this method provides significant benefit. Though enzymes with higher homology to the missing function are more likely to be the candidate to complete it, and we observed mostly enzymes above 40% homology completing the missing function, this is not the same as saying genes are readily annotated based on homology alone. With respect to the 40% homology annotation, we provide the following from Rost, *Journal of Molecular Biology*, 2002. Enzyme Function Less Conserved than Anticipated,

“...all groups substantially overestimated the conservation of enzyme function because their data sets were either too biased, or too small. An unbiased analysis suggested that less than 30% of the pair fragments above 50% sequence identity have entirely identical EC numbers...problems cannot be corrected easily by adjusting the thresholds for automatic transfer of genome annotations...”

In summary, enzymes that are 40% identical often have different functions. Moreover, as mentioned above, some genomes will have more than one open reading frame with >40% homology, complicating functional assignment.

Insert fragments were large. It remains unclear how often they contained a couple of genes instead of a single (based on gene predictions). Could the effect of an additional gene mask the completion of the auxotrophy? How many insert fragments actually need to be screened for an average genome size? Is the genomic DNA shearing optimal?

As mentioned in our response to Reviewer 1, with respect to how many genes are contained on a given fragment, here is a new plot showing the distribution of genes per fragment over all libraries (now Figure EV1),

As with Reviewer 1's question about the impact of operons, it is possible that a given gene's native context will impact its expression and subsequent identification in our assay. This is why we aim for a gene to be covered by more than one fragment. In addition, our data on the impact of the synthetic promoters (new Figure 4d) indicates that the synthetic promoter improves recovery of genes, which may overcome some of these issues.

With respect to the number of insert fragments that need to be screened, our aim was approximately 10x coverage of a genome. For example, the genome for *Acidovorax sp.* FHTAMBA is 4,752,000 bases in size. Breaking the genome into 3,000 bp fragments would require a minimum of 1,584 fragments (in the scenario where fragments were cut exactly at 3,000 bp intervals and spaced perfectly). As this is unrealistic, we aimed for 10x coverage. For *Acidovorax sp.* FHTAMBA, that meant a target of 15,840 fragments. Upon fragment library creation for this genome, we observed 18,608 mapped fragments. Some libraries fall above and below this 10x target, but this was the aim. Even in targeting 10x coverage, we observed that there simultaneously still genes with no coverage and genes covered by 10 or more fragments (new Figure EV2 as shown before),

Larger library sizes would have likely captured more genes, but would also have shifted this curve to the right, indicating greater redundancy (thus cloning inefficiency). In addition, it is possible that some percentage of the genes that are not covered fall into this category not because of random cloning loss but because of some unknown associated toxicity that selects for their non-inclusion.

With respect to the shearing method (sonication targeting 3 kb fragment size) – this physical method is generally unbiased, which is ideal. Our target size of 3 kb actually produces libraries with average fragment sizes between 1.9 and 2.8 kb (New Table EV2). The average size of a protein coding gene in our libraries ranges between 744 bp to 996 bp, and 92% - 98% of protein-coding genes are under 2.5 kb. Thus, to provide leeway for the genetic material before and after the protein coding gene (we cannot specify where the DNA is fragmented), having some “buffering” region before and after a gene is ideal. This mean fragment size across all libraries (2.2 kb) provides that. Moreover, our distribution of fragment sizes should allow for coverage of different size genes. Enzymatic shearing of DNA was also tested, but the yields of these processes provide too little DNA for subsequent cloning steps.

Bias to top candidates in liquid cultures depends on the growth rate difference between the clones. Perhaps the liquid growth assay could be performed if the maximum growth rate was reduced e.g., by the choice of the carbon source or other modification to the chemical environment.

Thank you for this suggestion. There is the possibility for future tuning of the liquid culture assays to improve recovery of a diversity of fragments/barcodes. That said, as mentioned in our response to Reviewer 1, even for just the case of the *E. coli* fragment library in $\Delta thrB$, the known “solution” BW25113_RS00015 (*thrB*) was contained on 15 different fragments (10 sense, 5 antisense), yet only one was observed in the assay. The scenario of *E. coli* fragments should represent the least expression bias of all cases, yet a single fragment still came to dominate the culture, meaning there is likely stochasticity to this process and even tweaking the growth rate may not fully alleviate this issue.

I would recommend removing the summary section from the results. It did not improve clarity and I found it repetitive.

We have moved this section to the Supplemental Materials as a Supplemental Note.

I have the following concerns on the formatting:
Past tense should be consistently used when reporting the work done.

Thank you, we have updated the manuscript accordingly.

Numbers up to twelve should be written out

We appreciate this suggestion and have made several corresponding changes to the manuscript. That said, we prefer to leave values such as >4, 3kb, and 1x as they are because we see them as more readable. We leave modifying these cases to the discretion of the editor, as we do not see a clear policy from *Molecular Systems Biology*.

Figure referencing calls for a revision like on page 6 the main text contains figure legend text.

The authors are unsure of this exact reference, but have checked for figure legend text in the main text and ensure that there are no cases of this.

Units for fitness are missing

As defined in the manuscript, fitness is a \log_2 change in the relative abundance, and thus is unitless. For the first figure in the manuscript that uses fitness, we have added “(\log_2 fold change)” to the y-axis label. For other figures in the main text that use fitness (Figure 3 and Figure 5), we have added a description to the caption.

"complimenting" in the abstract should be complementing

Thank you, we have corrected this.

Clarification for when statistical testing was used for confirming significance is occasionally missing.

We have updated the manuscript to include statistical information. The follow paragraph in the Expected hit Recovery section specifically has been updated,

“...When considering only inserts that cover the entire gene, were oriented correctly with respect to the synthetic promoter, and were detected in the t_0 , the mean RBS strength was 2.36 for inserts with significant benefits and 2.37 for those without ($P = 0.89$, t-test). Overall, the RBS strength prediction did not correlate with individual protein complementation success ($P = 0.40$, Wilcoxon rank sum test).

We also looked at genome GC content. Several of the genomes with low success rates have much higher GC content than *E. coli*. Across all inserts that contain an expected beneficial gene, are detected at t_0 , and in the sense orientation to the synthetic promoter, the median GC content was 61% for successful fragments and 66% for unsuccessful fragments ($P = 0.006$, Wilcoxon rank sum test). The poorer performance of the higher GC content genomes could have been related to expression via codon usage. However, comparing percentage of rare codons or codon adaptation index [CAI (Sharp & Li, 1987)] for genes that did or did not complement produced no clear takeaways. A modest correlation for CAI ($P = 0.049$, Wilcoxon rank sum test) was observed but in the opposite direction of expectation, meaning successful genes had a lower codon adaptation score (0.52) compared to unsuccessful genes (0.57), which is likely an artifact. **Appendix Figure S15** provides plots comparing successful and unsuccessful cases with respect to their GC content, predicted RBS strength, and codon adaptation index. As can be seen, successful and unsuccessful cases have almost completely overlapped values, and thus no obvious trend can be ascertained.”

Figure 3b *E. coli* should be *E. coli*

Thank you, we have fixed this.

In addition I have a couple of minor concerns:

By definition a minimal medium contains only the necessary nutrients for growth, is this the case here (page 4)?

Yes. This medium (M9, 1% glucose) only contains the necessary nutrients for growth. It contains relevant salts for osmolality, phosphorous, and nitrogen (osmolality - NaCl, 2.5 g/L; phosphorous - KH_2PO_4 , 15 g/L, Na_2HPO_4 , 33.9 g/L; nitrogen - NH_4Cl , 5 g/L), glucose as a carbon source, and magnesium and calcium (enzyme cofactors).

I would expect not knockouts but complementing genes on page 7.

We appreciate this comment. Our discussion starts with knockouts because that constrains what “answers” we could expect to find in our assay. From there, we move to describing, based on the source genomic material, what genes could be solutions to these knockouts.

Reviewer #3:

Manuscript ID: MSB-2024-12412

Title: High-throughput protein characterization by complementation using DNA barcoded fragment libraries

Authors: Bradley W. Biggs et al.

This manuscript describes the development of a new high-throughput barcoded plasmid system named "Coaux-Seq."

The authors first developed the expression vector with some modifications, including the barcode system to the original vector, pBbA2c-RFP(Addgene #35326), to create high-throughput shotgun expression libraries.

The authors conducted functional screening on 11 diverse bacterial genomic fragment libraries. They identified and functionally screened plasmids carrying functionally complementing fragments from 20 different auxotrophs of *E. coli* on the M9 minimal medium. The results of this screening provide a comprehensive understanding of the 'Coaux-Seq' system's capabilities.

This manuscript has valuable information for the readers and is acceptable for publication after carefully considering the comments below.

Major points;

1) The numerical information on the breakdown of candidate gene groups in the case studies "Expected recovery," "Diverged hits," and "Identifying novel enzymes and transporters" is very confusing. It would be easier to understand if it were shown in a flowchart or summarized in a simple table.

Thank you for this suggestion. We have now included a summary flow chart in the discussion (new Figure 9) shown below.

2) Is it unnecessary to examine the stability of the individual fragments in the established library? I

assume that the individual plasmids are not separated and purified but stored in a pool of plasmids, but if this is the case, it is also necessary to discuss how much amplification the cloned plasmids of individual fragments can withstand during the repeated use stage.

Moreover, in the current analysis, the authors primarily employed functional complementation. This approach, which relies on the similarity of the genes encoded in the complementary fragments for analysis and interpretation, can be viewed as a validation of the results of homology analysis. It is particularly intriguing as it has the potential to identify fragments encoding genes with very low homology or no homology, such as in convergent evolution. Could the authors delve deeper into such cases?

First, yes, the generated fragment libraries are stored as pools. Upon creation of the original libraries, we stored numerous glycerol stocks of the bacterial culture containing the original library, so that we can return to the library without going through freeze/thaw cyclers or retransformation. In addition, we miniprepmed approximately 200 μ L of the plasmid library before beginning complementation experiments. As only 5-7 μ L are used for each experiment, and with the plasmid material being stored at -20°C in the intervening period of time to aid in stability, we were able to use the originally prepped plasmid libraries for all experiments conducted in this study (no amplification). Plasmid libraries were stored in fresh nuclease free water. Under these conditions, plasmid libraries are generally stable for this period of time and we did not observe a loss in library efficiency over time.

The reviewer raises an interesting point about convergent evolution. It is true that complementation assays do not depend on homology. Indeed, we identified a TauE-like sulfate transporter (TK06_RS10770) that is not homologous to any previously-characterized sulfate transporter and is part of a different family compared to the permease components (CysW and CysT). It is possible that in this case we observed convergent evolution. We have added the following sentence to the manuscript in the section for this protein,

“Interestingly, as this protein does not bear homology to previously-characterized sulfate transporters and is in a different family compared to the permease components (CysW and CysT), it is possible that it represents an example of convergent evolution.”

Minor comments;

1) Page 4, line 18, "oriT" is mentioned in the text. p15A replication origin information is important information about the copy number of this vector, but what is the advantage of the oriT information for this purpose? If the authors have a clear, advantageous way of using oriT for the same purpose, I think it is generous to mention the benefit of oriT.

Thank you for this comment. First, we have moved this description to the methods section. The benefit of the oriT is to allow for the potential for conjugation, which was not used in this study, but we wanted to enable for future studies. In this work, we used electroporation for *E. coli* transformation. In the “General Cloning” section of the Methods, where this information has been relocated, we have added an explanation as to why the including oriT could be useful (i.e., conjugation).

2) Page 8, "Expected Hit Recovery" section, line 12, the definition of "t₀" is missing. "t₀" means the reads on the LB rich medium recovery. I think it should be clear in Figure 1 or the materials and methods part.

Thank you for raising this point. We have added a definition in this section (“Expected Hit Recovery”) and have added a t₀ to the appropriate location in Figure 1.

3) Page 5, Fig 2 a), I think there is no reason to show "not significance" by grey box and grey spots. Just

by grey spots, it might be simple. Is there any other reason to distinguish two ways? Or the authors need to add the statement (page 6, line 9, (fitness > 5 and z-like test statistic > 4)) to the legend.

We appreciate the reviewer's concern. The area of the box contains many (>4 million) spots that are not significant, which is difficult to visualize and take away meaning. We have left the region between 4 and 5 to show that this region lacks replicate consistency (showing what is just below our cutoff). We have updated the caption with the statement referenced by the reviewer to clarify what is indicated on the plot. In addition, we have added the following "Most pairs of measurements (>4 million) have no fitness benefit (\log_2 ratio <4) and lie within the gray box."

4) Page 8, "Expected Hit Recovery," 13, "remaining 152 genes covered by a fragment, only 132 of them were seen at t_0 ." why "remaining"? Out of 203 expected complementing genes, 75% means 152 genes, I read. Is this correct? If so, I think "remaining" is unreasonable. Out of 152, 132 were seen by sequence reads at t_0 . So, 20 fragments were not observed at t_0 . This depends on the number of long-read identifications of the correspondence between the fragments and barcodes, the size of the initial library, and the total genome size of 11 strains. What is the estimated coverage of the library of the genomes analyzed? This situation might be reasonable if functional screening were used. "2% vs. 50%" is also confusing for me. These numbers came from where?

First, yes, 152 "remaining" represent the 75% of the 203 genes that were successfully cloned. Essentially the paragraph reads as a set of filters. We have created a new flowchart figure (Figure 9 shown above), which may help clarify. This logic flows that – starting from the genomes themselves, there were 203 genes we could reasonably expect to complete missing function in our assays. Only 152 of those genes were observed in a cloned fragment. The other 51 were not. When we next speak of observing fragments at t_0 , only those 152 could possibly be observed, thus we label them the "remaining" of the 203. To you next question, again, yes 20 of the fragments were not observed at t_0 . Thus, 132 remain after that filtering step.

The estimated coverage of the libraries for the genomes analyzed can be found in Table EV1 in the last column on the right.

With respect to the 2% and 50% numbers, these represent the proportion of fragments that were successful if they were detected at t_0 or not. Meaning, if a fragment contained a complete gene that should provide a benefit and was observed at t_0 , it did provide a benefit 50% of the time. Alternatively, if a known fragment contained a full gene that should provide a benefit but was not detected at t_0 , it provided a benefit 2% of the time. We have updated the text to clarify this to say,

"Moreover, as would be expected, inserts that contain a should-be beneficial protein but were not detected in the t_0 samples (0 reads for their barcode) are much less likely to show a benefit compared to inserts that are detected. If a fragment contained a complete gene that was expected to provide and was observed at t_0 , then we observed a benefit 50% of the time. If a fragment contained a complete gene that was expected to provide a benefit but was not observed at t_0 , then we observed a benefit just 2% of the time, with the 2% likely representing fragments that were transformed at low abundance and missed in the t_0 sequencing."

5) Page 9, line 8, and supplementary information Fig. S15, in the main text, shows 72% but in the Sup. Fig. 15 shows the ratio. Unifying these units is clearer.

Both were originally shown as fractions. However, we have moved Figure S15 into the main text as part of the new Figure 4 (Figure 4d). In the new configuration all are shown as fractions, which should help clarify.

6) Page 9, line 11, and supplementary information Fig. S15, species names in the Fig. S15 X-axis and the main text are different. Unifying those names is easier.

Thank you for this suggestion. We have updated to have a single legend for Figure 4. The order of the strains is consistent throughout now.

7) Page 9, second paragraph line 26, "poorly (35% recovery)", it is not easy for readers to identify :S, koreensis" on Fig. S15.

We appreciate the challenge that the reviewer highlights. With updated Figure 4, we hope this is now much more readily observed.

8) Page 9, second paragraph line 32, "Overall, the RBS strength prediction did not correlate with individual protein complementation success.", I think the plasmid is recoverable and is it possible to check the expression individually?

While this approach is possible, a thorough test would require examining a suitable number of cases and from many different strains and for fragments that did and didn't work to create a statistically reliable, generalized trend. Otherwise, we are left testing each case individually and getting "one off" answers, which is counter to the high-throughput aim. Based on the data that we show in the newly made Figure S15, we would not expect a trend based on RBS function. Because the synthetic promoter is providing high levels of transcription, even weak or "spurious" RBS activity should be selected for in a functional complementation assay like ours (strong selection pressure). Testing a fluorescent reporter in the place of the gene would provide only an approximation (as it cannot be tested in auxotrophic background under the minimal medium conditions and because RBS function is known to be dependent upon genetic context – Mutalik et al., *Nature Methods*, 2013). In the scenario where we observe the fluorescent reporter in place of a gene that did not complement, we would still not know why complementation was not successful (perhaps the enzyme just performs poorly in *E. coli*). In the case where we don't observe expression, it would indicate expression is likely the underlying issue, which would either lead to individual cloning (no longer high-throughput) or a changing of host system for selection, which is one of next planned set of experiments for this method.

9) Page 10, "Diverged hits" section, "Additionally, we categorized a set of "diverged" hits (10 in total) that could have been expected to complete the missing function, but with caveats to high-confidence assignment. Hits were considered "diverged" if they were similar to experimentally-characterized proteins that have the missing activity (and not more similar to proteins known to have other functions instead), but are <40% identical on an amino acid basis to any characterized protein with that activity in curated databases". For me, it is not easy to understand. "similar" means structural similarity by sequence similarity analysis? If so, "<40%" means not a similar sequence, probably in the amino acids sequence. Is this correct? This situation seems interesting, and it might be possible to consider this as the case of convergent evolution. Why didn't you try to compare the estimated 3D structure by AlphaFold? If these connect to Figs. 4 and 5, re-arrange the "Diverged hits" and "Identifying novel enzymes and transporters" sections.

We have updated the text to say "...<40% identical on an amino acid sequence basis..." to help clarify this. We are comparing amino acid identity and not structure. Sequences that are sufficiently similar for their homology to be detected by BLAST virtually always have very similar structures, especially at >30% identity (Chothia and Lesk, *EMBO Journal*, 1986). The relation between the divergence of sequence and structure in proteins). Thus, we would not expect predicted structure comparisons to be helpful in these cases. As a single example, using the crystal structures for *E. coli* metB (PDB 1CS1) and metC (PDB 1CL1)

and aligning in PyMOL, one observes a very similar structure (RMSD = 1.501), yet these enzymes have entirely distinct roles. Image of overlapping regions below.

10) Page 10, "Identifying novel enzymes and transporters" section, line 10, "cross feeding" and Sup. Figs 31-33, can the secreted metabolites reach the neighboring spots even at such a distance? This is very interesting, but can we call it cross-feeding? It is possible to mix isolated strains and see the growth of each cell more quantitatively, but is this not considered necessary?

Yes, the secreted metabolite can reach neighboring spot. The molecules and their diffusivity in agar are comparable to that of antibiotics, which are well known to behave in this manner on agar plates (As an example, Figure 1 from Fleming, *British Journal of Experimental Pathology*, 1929, On the Antibacterial Action of Cultures of *Penicillium*, with Special Reference to their Use in the Isolation of *B. influenzae*). The clear distinction on plates is an easier way to observe this.

11) Page 13 last line, "Moreover, RB-TnSeq data for LRK54_RS05660 does not show auxotrophic phenotypes" is not surprising because of the redundancy of genes. Did you check the genes in the same strain that are similar?

We suspect that a different protein, LRK54_RS05305, is the metB or metZ of *Rhodanobacter denitrificans* FW104-10B01: it is 59% identical to *E. coli*'s metB and has a very similar fitness pattern as metA (LRK54_RS05310, $r = 0.92$, Adam Deutschbauer, personal communication). Unfortunately, our library did not contain any inserts that covered all of LRK54_RS05305.

12) Page 14, line 3 and 5, LRK54_RS05660 is the target to be analyzed for the complementation to delta_metC but LRK54_RS17455 is mentioned in line 5. Is this just a mistake or my misunderstanding?

This was a typo, thank you for catching it.

13) On page 14, line 19, I think this point is important. Is there any structure-based discussion about this point? Is there a difference at the catalytic active center or other structural domains for other functional possibilities?

As referenced above, the PLP-dependent enzymes metB and metC are very similar. They have similar active sites with a catalytic lysine, an arginine that increases the nucleophilicity of a tyrosine, and a tyrosine that abstracts a proton from the substrate (please see <https://www.ebi.ac.uk/thornton-srv/m-csa/entry/759/>; <https://www.ebi.ac.uk/thornton-srv/m-csa/entry/449/>). Other related enzymes, such as methionine gamma-lyase, have similar active sites. LRK54_RS05660 has all of these residues.

To the best of our knowledge, the mechanism of MetZ has not been studied, but it is likely a PLP-dependent enzyme as well (i.e. the key lysine is conserved). However, the tyrosine that is proposed to act as an acid/base catalyst has become a phenylalanine. Because of the highly overlapping structural similarities among metB, metC, and metZ, structure isn't particularly helpful for differentiation.

14) Page 15, line 7, "Coaux-Seq successfully uncovered novel biochemical function." I think it is generally understood that functional complementation fragment analysis is an essential element and that barcode technology enables quantitative analysis of this using a pooled library. However, I feel a little uncomfortable when it is vigorously explained that Coaux-Seq can do it.

Coaux-Seq is fundamentally a combination of traditional complementation assays with the high-throughput analysis enabled next generation sequencing (BarSeq). We hope that our new line in the discussion seen below helps to clarify this.

"Similar to how RB-TnSeq dramatically improved the throughput of transposon-based loss-of-function experiments that have been employed for many years (Hensel *et al*, 1995; Cain *et al*, 2020), Coaux-Seq significantly improves the throughput of gene complementation studies, which have been utilized for even longer (Fincham, 1968)."

26th Aug 2024

Manuscript Number: MSB-2024-12412R

Title: High-throughput protein characterization by complementation using DNA barcoded fragment libraries

Author: Bradley Biggs

Morgan Price

Dexter Lai

Jasmine Escobedo

Yuridia Fortanel

Yolanda Huang

Kyoungmin Kim

Valentine Trotter

Jennifer Kuehl

Lauren Lui

Romy Chakraborty

Adam Deutschbauer

Adam Arkin

Dear Adam,

Thank you for sending us your revised manuscript. We have now heard back from the two reviewers who agreed to evaluate your revised study. As you will see below, the reviewers are satisfied with the performed revisions and support publication. Before we can formally accept the manuscript for publication, we would ask you to address some remaining issues listed below.

1. Please do not include the figures in the manuscript file.
2. Please upload the Reagents and Tools table separately.
3. Please remove the Author Contribution section from the manuscript text.
4. The manuscript sections should be in the following order: Title page - Abstract & Keywords - Introduction - Results - Discussion - Methods - Data Availability - Acknowledgments - Disclosure Statement & Competing Interests - References - Figure Legends - (Main Tables with legends) - Expanded View Figure Legends.
5. Funding information: Only one funder was entered in the submission system (DOE DE-AC02-05CH11231). Please add the rest of the funders as separate funder entries (not in the Comments box): Dean A. Richard Newton Memorial Professor Chair Funds; ENIGMA - Ecosystems and Networks Integrated with Genes and Molecular Assemblies; Science Focus Area Program at Lawrence Berkeley National Laboratory; QB3 Genomics, University of California Berkeley (Berkeley, CA)- RRID:SCR_022170; NIH S10 OD018174 Instrumentation Grant.
6. Please add callouts for individual panels of Figure 1.
7. Data availability: please provide specific URLs for CP151802, CP151803, CP152408, CP152407 and CP152376 datasets in the data availability statement.
8. Individual EV figures need to be uploaded separately and should not be provided in the manuscript file.
9. EV datasets
 - the separate file with the legends needs to be removed;
 - each legend should be provided in its corresponding Excel file as a separate tab/sheet;
 - Since Dataset EV4 is not so complex, it should be uploaded as an EV Table (instead of EV dataset). Please update the callouts accordingly.
 - The 2 EV Tables need to be removed from the manuscript file together with the legends, and uploaded as separate files; if you intend to keep them in the manuscript file, then these should be renamed to Table 1 and Table 2.
10. Appendix:
 - "Supplemental" should be renamed to "Appendix";
 - The correct nomenclature for the figures and tables should be "Appendix Figure S1", etc. "Appendix Table S1" etc.; Callouts also need to be updated with "Appendix" word where necessary;
 - The references in the Appendix need to be in the same format as in the manuscript file.

11. I have slightly modified and shortened the standfirst text (see attached). Please let me know if it's fine as is or if you would like to introduce further modifications.

12. Please address the following issues raised by our data editor:

- Please note that the legend for figure EV2 is provided in the manuscript, however the figure for the same is missing. Additionally, there is no figure labelled as EV 1 provided in the manuscript, we are not sure if the expanded view figures are missing or there is a labelling issue in the manuscript. Kindly look into this.
- Please note that the measure of center for the error bar needs to be defined in the legend of figure 4d.
- Please note that axis label is not defined for figure 4d.

When you resubmit your manuscript, please download our CHECKLIST (<https://bit.ly/EMBOPressAuthorChecklist>) and include the completed form in your submission. *Please note* that the Author Checklist will be published alongside the paper as part of the transparent process (<https://www.embopress.org/page/journal/17444292/authorguide#transparentprocess>)

Click on the link below to submit your revised paper.

Kind regards,
Jingyi

Jingyi Hou, PhD
Scientific Editor
Molecular Systems Biology

Reviewer #1:

I am satisfied by the authors' response.

Reviewer #2:

The Authors have addressed all my concerns in the revision. I recommend publication of the manuscript.

All editorial and formatting issues were resolved by the authors.

9th Sep 2024

Manuscript number: MSB-2024-12412RR

Title: High-throughput protein characterization by complementation using DNA barcoded fragment libraries

Dear Adam,

Thank you again for sending us your revised manuscript. I am pleased to inform you that your paper has been accepted for publication.

Before we export the manuscript to our production team, could you please let us know if you are fine with the modified standfirst text (see attached)?

Your manuscript will then be processed for publication by EMBO Press. It will be copy edited and you will receive page proofs prior to publication. Please note that you will be contacted by Springer Nature Author Services to complete licensing and payment information.

Kind regards,
Jingyi

Jingyi Hou, PhD
Scientific Editor
Molecular Systems Biology
